# VIKING: Deep variational inference with stochastic projections

**Samuel G. Fadel, Hrittik Roy, Nicholas Krämer,**
**Yevgen Zainchkovskyy, Stas Syrota, Alejandro Valverde Mahou**
Technical University of Denmark
{samma,hroy,pekra,yeza,stasy,avama}@dtu.dk

**Carl Henrik Ek**
University of Cambridge
che29@cam.ac.uk

**Søren Hauberg**
Technical University of Denmark
sohau@dtu.dk

## Abstract

Variational mean field approximations tend to struggle with contemporary over-parametrized deep neural networks. Where a Bayesian treatment is usually associated with high-quality predictions and uncertainties, the practical reality has been the opposite, with unstable training, poor predictive power, and subpar calibration. Building upon recent work on reparametrizations of neural networks, we propose a simple variational family that considers two independent linear subspaces of the parameter space. These represent functional changes inside and outside the support of training data. This allows us to build a fully-correlated approximate posterior reflecting the overparametrization that tunes easy-to-interpret hyperparameters. We develop scalable numerical routines that maximize the associated evidence lower bound (ELBO) and sample from the approximate posterior. Empirically, we observe state-of-the-art performance across tasks, models, and datasets compared to a wide array of baseline methods. Our results show that approximate Bayesian inference applied to deep neural networks is far from a lost cause when constructing inference mechanisms that reflect the geometry of reparametrizations.

## 1 The troubles of the overparametrized Bayesian

Bayesian inference provides an attractive framework for learning from data, marginalizing a data-dependent likelihood with a data-independent prior leads to a combined probability measure that can be used for learning. For most model classes, the marginalization is intractable, and we resort to approximate methods of integration. While this has shown itself to be a very successful approach, overparametrized models have subtle characteristics that lead to significant challenges in practice.

Specifically, overparametrizations lead to a many-to-one relationship where several parameter configurations describe the same function. Dinh et al. (2017) exemplify that the neural network $f(x) = w_1 \operatorname{ReLU}(w_2 x)$ can be reparametrized as $f(x) = {w_1}/{\gamma} \operatorname{ReLU}(\gamma w_2 x)$ for any $\gamma \neq 0$, implying that the weights $(w_1, w_2)$ and $({w_1}/{\gamma}, \gamma w_2)$ yield *identical functions*. Thus, despite having two trainable parameters, the neural network only has one degree of freedom. Making matters worse, the discrepancy between the number of parameters and degrees of freedom generally grows with the model size. If the marginalization can be performed analytically, the resulting measure will reflect this structure; however, if we need to resort to approximate methods, we need to carefully design our approximation so that the correct geometry of the problem can be parametrized. If not, the resulting object will not be a valid probability measure over the space of functions and application of this

39th Conference on Neural Information Processing Systems (NeurIPS 2025).

object will lead to pathological solutions where the same function has different densities dependent on its parametrization (Watanabe, 2009).

In recent work, Roy et al. (2024) showed that the parameters describing the same function are continuously connected sets in the weight space. This has damaging effects as traditional posterior approximations like Laplace approximations (MacKay, 1992) or mean-field approximations (Blundell et al., 2015) cannot accurately reflect the reparametrizations, which lead to the disappointing performance of approximate Bayesian inference in these types of models.

**In this paper**, we study variational inference applied to overparametrized deep neural networks. We propose an approach that takes the intractable posterior and factorizes it into two orthogonal parts, a data-dependent and a data-independent measure. These can, intuitively, be seen as modeling uncertainties on the training data in one subspace and elsewhere in the other. Importantly, this leads to an easy-to-interpret structure analogous to the likelihood and the prior that lead to the posterior in the first place. Despite having an interpretable and simple decomposition, our approach is fundamentally fully correlated and captures dependencies between all model parameters, addressing the pathological behavior of previous methods. This introduces a computational challenge, which we address through a stochastic extension of the alternating projections algorithm that was recently proposed for *post hoc* posterior approximation (Miani et al., 2025). The result is a scalable, architecture-agnostic approach to Bayesian training of deep neural networks that achieves state-of-the-art performance on a wide array of contemporary models and datasets. The both code for reproducing our experiments[1] and an easy-to-use library[2] are available online.

## 2 Background and related work

The aspiration of Bayesian deep learning is to combine the predictive advantages of deep neural networks with the theoretical justifications of the Bayesian framework, hoping to achieve the benefit of both. While the individual layers in a neural network are simple mathematical objects, the compositional structure of the layers renders posterior inference analytically intractable for any interesting model. There has been a range of different approaches to recover posterior estimates of the neural network weights (Hernandez-Lobato and Adams, 2015; Hron et al., 2018; Li et al., 2015; MacKay, 1995), but often the benefits have failed to materialise, some even restricting the Bayesian treatment to a subset of parameters (Zhao et al., 2024). Rather, it is common to experience either *over-* or *underfitting* (Daxberger et al., 2021; Kristiadi et al., 2022; Wenzel et al., 2020; Zhang et al., 2024), and most Bayesian training schemes are reported to be brittle (Warburg et al., 2023). Success in achieving the promises of a Bayesian treatment of deep neural networks has been so elusive that it raises the question if there is something inherent in their structure that renders it numb to a Bayesian approach (Sharma et al., 2023).

Singular learning theory (Watanabe, 2009) argues that the troubling structure comes from over-parametrizations, where a single function can be described in more than one way. While mathematically elegant, the algebraic geometric approach has been difficult to turn into computational practice. Instead, Roy et al. (2024) framed the problem in terms of differential geometry by studying how the geometry of the function space embeds itself into the parameter space. From this, the authors were able to characterize the manifold of reparametrizations in a manner that lends itself to computation. Specifically, the manifold can be characterized locally by the kernel[3] of the Fisher–Rao metric over the network parameters $\boldsymbol{\theta}$ (Amari, 2016),

$$\mathcal{F}_{\boldsymbol{\theta}} = \mathbb{E}_{\mathbf{y} \sim p_{\boldsymbol{\theta}}(\mathbf{y}|\mathbf{x})} \left[ \frac{\partial \log p_{\boldsymbol{\theta}}(\mathbf{y}|\mathbf{x})}{\partial \boldsymbol{\theta}} \frac{\partial \log p_{\boldsymbol{\theta}}(\mathbf{y}|\mathbf{x})}{\partial \boldsymbol{\theta}}^{\top} \right] \in \mathbb{R}^{D \times D}. \tag{1}$$

Here $\log p_{\boldsymbol{\theta}}(\mathbf{y}|\mathbf{x})$ is the log-likelihood of the data parametrized by $\boldsymbol{\theta} \in \mathbb{R}^{D}$. The kernel of this matrix infinitesimally gives directions in weight space where $\log p_{\boldsymbol{\theta}}(\mathbf{y}|\mathbf{x})$ is constant for all observations in the support of the data. This provides a characterization of why Bayesian learning has not succeeded, showing that for deep learning to harvest the benefit of a Bayesian treatment, the geometrical structure needs to be reflected in the approximate posterior. For a detailed discussion, we refer to (Kunstner et al., 2019; Miani et al., 2025).

---

[1]`https://github.com/eugene/viking-paper-experiments`
[2]`https://github.com/fadel/viking`
[3]Throughout the paper, *kernel* refers to the *null space* of a matrix, while *image* is its orthogonal complement.

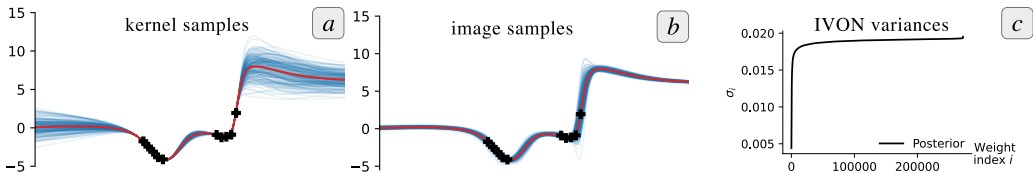

Figure 1: Panels *a* and *b* show isotropic Gaussian samples that have been projected onto the *kernel* and *image* of the empirical Fisher–Rao metric associated with a neural network trained on the shown data. The kernel samples retain the predictions of the neural network, while the image samples do not. Panel *c* shows the learned weight-variances of IVON trained on CIFAR-10. These variances are near-identical across weights, suggesting that an isotropic approximate posterior has been learned.

Miani et al. (2025) take steps towards integrating knowledge of overparametrization into a *post-hoc* approximate posterior. They note that the kernel of per-datum loss-Jacobian, which is closely related to the Fisher–Rao metric (1), consists of functions with identical training loss. Assuming noise-free data, they restrict the approximate posterior to only have probability mass in this kernel, resulting in a posterior approximation from which all samples have the same loss (see Figure 1a,b).

**Variational inference** (Blei et al., 2017) has shown itself to be an efficient and robust approach to approximate Bayesian inference and has been applied in several different ways to deep neural networks. Existing variational approximations predominantly use Gaussian variational distributions $p(\boldsymbol{\theta}|\mathbf{x}) \approx \mathcal{N}(\boldsymbol{\theta}|\hat{\boldsymbol{\theta}}, \boldsymbol{\Sigma})$, see, e.g., (Blundell et al., 2015; Shen et al., 2024), where $\hat{\boldsymbol{\theta}}$ and $\boldsymbol{\Sigma}$ are the variational parameters learned. To ensure that the approximation is tractable, the covariance $\boldsymbol{\Sigma}$ is commonly given a diagonal or block-diagonal structure, i.e., a *mean field* approximation (Daxberger et al., 2021). Current state-of-the-art in this direction appears to be *IVON* (Shen et al., 2024), which develops a Newton-like optimization scheme for the *evidence lower bound (ELBO)* associated with the mean field approximation. In our experience, IVON tends to learn near-identical variances for all network weights (Figure 1c), suggesting that the posterior approximation is inaccurate. Further, mean field approximations tend to be brittle and sensitive to the choice of hyperparameters. One hypothesis for this behavior is that the independence assumptions of mean field cause instabilities in overparametrized models, where parameters, by construction, are highly correlated. In this work, we want to build a full variational inference method that respects the geometry induced by the overparametrization characterized by Roy et al. (2024). To achieve this, we split the parameter space into the kernel and image of the Fisher–Rao metric, considering these independent.

**The numerical computations** associated with non-trivial covariance structures can be daunting. The first challenge is that for large neural networks, the covariance matrix cannot be stored in memory as its size grows quadratically with the number of network parameters. One remedy is to note that some matrices, e.g., the Fisher–Rao metric (1) and the generalized Gauss–Newton matrix (Daxberger et al., 2021), are functions of the training data. This implies that vector products of such matrices can be implemented without instantiation — a so-called *matrix-free* approach. This allows for working with non-trivial covariances at the cost of a higher compute budget, see, e.g., (Miani et al., 2024, 2025; Roy et al., 2024) for examples.

## 3  VIKING: Variational Inference with Kernel- and Image-spaces of numerical Gauss–Newton matrices

We next build a variational training loop designed with overparametrization in mind. We start from the ever-present prior $p(\boldsymbol{\theta}) = \mathcal{N}(\boldsymbol{\theta}|\mathbf{0}, \alpha^{-1}\mathbb{I})$ over parameters $\boldsymbol{\theta} \in \mathbb{R}^D$, with precision $\alpha \in \mathbb{R}$. As motivated earlier, we choose a variational family $q(\boldsymbol{\theta})$ that is linked to the kernel of an empirical estimate of the Fisher–Rao (FR) metric at $\hat{\boldsymbol{\theta}}$, denoted $\ker(\mathcal{F}_{\hat{\boldsymbol{\theta}}}) \subset \mathbb{R}^D$. Specifically,

$$q(\boldsymbol{\theta}) = \mathcal{N}(\boldsymbol{\theta}|\hat{\boldsymbol{\theta}}, \boldsymbol{\Sigma}_{\hat{\boldsymbol{\theta}}}), \quad \boldsymbol{\Sigma}_{\hat{\boldsymbol{\theta}}} = \sigma_{\text{ker}}^2 \mathbf{U}_{\hat{\boldsymbol{\theta}}} \mathbf{U}_{\hat{\boldsymbol{\theta}}}^\top + \sigma_{\text{im}}^2 (\mathbb{I} - \mathbf{U}_{\hat{\boldsymbol{\theta}}} \mathbf{U}_{\hat{\boldsymbol{\theta}}}^\top). \tag{2}$$

Here, $\mathbf{U}_{\hat{\boldsymbol{\theta}}} \in \mathbb{R}^{D \times R}$ is a $R$-dimensional basis of the kernel, such that $\mathbf{U}_{\hat{\boldsymbol{\theta}}} \mathbf{U}_{\hat{\boldsymbol{\theta}}}^\top$ is a projection matrix that maps parameter vectors in $\mathbb{R}^D$ to the linear subspace $\ker(\mathcal{F}_{\hat{\boldsymbol{\theta}}})$. In other words, for any $\mathbf{v} \in \mathbb{R}^D$ we have $\mathbf{U}_{\hat{\boldsymbol{\theta}}} \mathbf{U}_{\hat{\boldsymbol{\theta}}}^\top \mathbf{v} \in \ker(\mathcal{F}_{\hat{\boldsymbol{\theta}}})$, wherein the model's loss is unchanged at the data used to estimate the FR.

**Algorithm 1:** The VIKING algorithm

---

**Input:** Initial values of $(\hat{\boldsymbol{\theta}}, \sigma_{\mathrm{ker}}, \sigma_{\mathrm{im}})$, dataset with $B$ mini-batches
**Output:** Optimized values of $(\hat{\boldsymbol{\theta}}, \sigma_{\mathrm{ker}}, \sigma_{\mathrm{im}})$, representing $q(\boldsymbol{\theta})$ (Equation 2)

**1 for** each epoch **do**
**2**     **for** each Monte Carlo sample $s = 1$ **to** $S$ **do**
**3**         Take $\boldsymbol{\epsilon}^{(s,0)} \sim \mathcal{N}(\mathbf{0}, \mathbb{I})$
**4**         **for** each mini-batch $t = 1$ **to** $B$ **do**
**5**             Compute $\boldsymbol{\epsilon}^{(s,t)}$ from $\boldsymbol{\epsilon}^{(s,t-1)}$ (Equation 14)          ▷ *Alternating projections*
**6**         $\boldsymbol{\epsilon}_{\mathrm{ker}}^{(s,0)} \leftarrow \boldsymbol{\epsilon}^{(s,B)}$
**7**     **for** each mini-batch $t = 1$ **to** $B$ **do**
**8**         **for** each Monte Carlo sample $s = 1$ **to** $S$ **do**
**9**             Compute $\boldsymbol{\epsilon}_{\mathrm{ker}}^{(s,t)}$ from $\boldsymbol{\epsilon}_{\mathrm{ker}}^{(s,t-1)}$ (Equation 18)
**10**             $\boldsymbol{\epsilon}_{\mathrm{im}}^{(s,t)} \leftarrow \boldsymbol{\epsilon}^{(s,0)} - \boldsymbol{\epsilon}_{\mathrm{ker}}^{(s,t)}$
**11**             $\boldsymbol{\theta}^{(s)} \leftarrow \hat{\boldsymbol{\theta}} + \sigma_{\mathrm{ker}}\boldsymbol{\epsilon}_{\mathrm{ker}}^{(s,t)} + \sigma_{\mathrm{im}}\boldsymbol{\epsilon}_{\mathrm{im}}^{(s,t)}$          ▷ *A sample from $q(\boldsymbol{\theta})$*
**12**         $\hat{\boldsymbol{\theta}}, \sigma_{\mathrm{ker}}, \sigma_{\mathrm{im}} \leftarrow$ Gradient step on $\mathcal{L}(\hat{\boldsymbol{\theta}}, \sigma_{\mathrm{ker}}, \sigma_{\mathrm{im}})$ using $\{\boldsymbol{\theta}^{(s)}\}_{s=1}^{S}$ (Equation 3)

---

This proposed approximate posterior is perhaps the simplest possible choice that is adapted to overparametrized models: it has one scalar parameter, $\sigma_{\mathrm{im}}^2 \in \mathbb{R}_+$, that influences the predictive uncertainty on the training data, and another scalar parameter $\sigma_{\mathrm{ker}}^2 \in \mathbb{R}_+$ that reflects uncertainty elsewhere. Arguably, it is overly simplistic to calibrate a neural network through two scalars, but we will see that it is surprisingly competitive, suggesting that explicitly reflecting overparametrization is important in Bayesian approximations.

### 3.1 The VIKING ELBO

To estimate $q(\boldsymbol{\theta})$, we maximize the usual (variational) lower bound $\mathcal{L}$ of the likelihood $p(\mathbf{y}|\boldsymbol{\theta}, \mathbf{x})$

$$\mathcal{L}(\hat{\boldsymbol{\theta}}, \sigma_{\mathrm{ker}}, \sigma_{\mathrm{im}}) = \mathbb{E}_{\boldsymbol{\theta} \sim q}\left[\log p(\mathbf{y}|\boldsymbol{\theta}, \mathbf{x})\right] - \mathrm{KL}(q(\boldsymbol{\theta})\|p(\boldsymbol{\theta})), \tag{3}$$

where KL denotes the Kullback-Leibler divergence. We call the result *Variational Inference with Kernel- and Image-spaces of numerical Gauss–Newton matrices (VIKING)*. The overall algorithmic steps involved in our method are summarized in Algorithm 1. We refer to the two terms of the ELBO as the *reconstruction term* and the *KL term*, respectively, and discuss their evaluation in what follows.

**The reconstruction term.** Assume access to an algorithm to approximate products between the projection matrix $\mathbf{U}_{\hat{\boldsymbol{\theta}}}\mathbf{U}_{\hat{\boldsymbol{\theta}}}^{\top}$ and any vector $\mathbf{v} \in \mathbb{R}^D$. For $s = 1, \ldots, S$, we can then estimate of the reconstruction term as

$$\boldsymbol{\epsilon}^{(s)} \sim \mathcal{N}(0, \mathbb{I}) \qquad\qquad \in \mathbb{R}^D, \tag{4}$$

$$\text{(projection onto kernel space)} \quad \boldsymbol{\epsilon}_{\mathrm{ker}}^{(s)} = \mathbf{U}_{\hat{\boldsymbol{\theta}}}\mathbf{U}_{\hat{\boldsymbol{\theta}}}^{\top}\boldsymbol{\epsilon}^{(s)} \qquad\qquad \in \mathbb{R}^D, \tag{5}$$

$$\text{(image space is orthogonal)} \quad \boldsymbol{\epsilon}_{\mathrm{im}}^{(s)} = (\mathbb{I} - \mathbf{U}_{\hat{\boldsymbol{\theta}}}\mathbf{U}_{\hat{\boldsymbol{\theta}}}^{\top})\boldsymbol{\epsilon}^{(s)} = \boldsymbol{\epsilon}^{(s)} - \boldsymbol{\epsilon}_{\mathrm{ker}}^{(s)} \qquad\qquad \in \mathbb{R}^D, \tag{6}$$

$$\boldsymbol{\theta}^{(s)} = \hat{\boldsymbol{\theta}} + \sigma_{\mathrm{ker}}\boldsymbol{\epsilon}_{\mathrm{ker}}^{(s)} + \sigma_{\mathrm{im}}\boldsymbol{\epsilon}_{\mathrm{im}}^{(s)} \qquad\qquad \in \mathbb{R}^D, \tag{7}$$

$$\mathbb{E}_{\boldsymbol{\theta} \sim q}\left[\log p(\mathbf{y}|\boldsymbol{\theta}, \mathbf{x})\right] \approx \frac{1}{S}\sum_{s=1}^{S} \log p(\mathbf{y}|\boldsymbol{\theta}^{(s)}, \mathbf{x}) \qquad\qquad \in \mathbb{R} \; . \tag{8}$$

Here, Equation 5 requires careful computation, to be addressed later. Once $\boldsymbol{\epsilon}_{\mathrm{ker}}^{(s)}$ is computed, obtaining a sample $\boldsymbol{\theta}^{(s)}$ from the approximate posterior is trivial.

**The KL term.** Since we have both a Gaussian prior and a Gaussian approximate posterior, we can evaluate the KL term in closed form with

$$\mathrm{KL}(q(\boldsymbol{\theta})\|p(\boldsymbol{\theta})) = \frac{1}{2}\left(\alpha\mathrm{Tr}(\boldsymbol{\Sigma}_{\hat{\boldsymbol{\theta}}}) - D + \alpha\|\hat{\boldsymbol{\theta}}\|^2 - D\log(\alpha) - \log\det(\boldsymbol{\Sigma}_{\hat{\boldsymbol{\theta}}})\right). \tag{9}$$

Recall the construction of $\mathbf{\Sigma}_{\hat{\boldsymbol{\theta}}}$ from Equation 2. Since $\mathbf{\Sigma}_{\hat{\boldsymbol{\theta}}}$ is a sum of scaled projection matrices, it has $R$ eigenvalues equal to $\sigma_{\mathrm{ker}}^2$ and $D-R$ eigenvalues equal to $\sigma_{\mathrm{im}}^2$. We can, thus, evaluate

$$\mathrm{Tr}(\mathbf{\Sigma}_{\hat{\boldsymbol{\theta}}}) = \sigma_{\mathrm{ker}}^2 R + \sigma_{\mathrm{im}}^2(D-R), \text{ and} \tag{10}$$

$$\log\det(\mathbf{\Sigma}_{\hat{\boldsymbol{\theta}}}) = 2R\log(\sigma_{\mathrm{ker}}) + 2(D-R)\log(\sigma_{\mathrm{im}}), \tag{11}$$

requiring only an estimate of the kernel dimension $R$ to evaluate the full KL term in closed form.

To estimate $R$, we note that $\mathbf{U}_{\hat{\boldsymbol{\theta}}}\mathbf{U}_{\hat{\boldsymbol{\theta}}}^\top$ is a projection matrix, i.e., it has $R$ eigenvalues that are 1 while the remaining are 0. Consequently,

$$R = \mathrm{Tr}(\mathbf{U}_{\hat{\boldsymbol{\theta}}}\mathbf{U}_{\hat{\boldsymbol{\theta}}}^\top) \approx \frac{1}{S}\sum_{s=1}^{S}\boldsymbol{\epsilon}^{(s)\top}\overbrace{\mathbf{U}_{\hat{\boldsymbol{\theta}}}\mathbf{U}_{\hat{\boldsymbol{\theta}}}^\top\boldsymbol{\epsilon}^{(s)}} = \frac{1}{S}\sum_{s=1}^{S}\boldsymbol{\epsilon}^{(s)\top}\boldsymbol{\epsilon}_{\mathrm{ker}}^{(s)}, \tag{12}$$

where the approximation is Hutchinson's trace estimator (Hutchinson, 1989). Note that we may reuse $\boldsymbol{\epsilon}_{\mathrm{ker}}^{(s)}$ (Equation 5) from the reconstruction term.

## 3.2 Kernel projections

At first glance, the ELBO terms (Equations 8 and 9) seem straightforward. The practical difficulty lies in projecting onto $\mathrm{ker}(\mathcal{F}_{\hat{\boldsymbol{\theta}}})$ as per Equation 5. One noticeable challenge is $\mathbf{U}_{\hat{\boldsymbol{\theta}}}\mathbf{U}_{\hat{\boldsymbol{\theta}}}^\top$ having $D\times D$ entries, where $D$ is the number of model parameters. With many modern models having millions or billions of parameters, it is not possible to instantiate this matrix, and we resort to matrix-free algorithms.

We start by defining the matrix containing a stack of *per-datum* loss gradients,

$$\mathbf{J}_{\hat{\boldsymbol{\theta}}} = \begin{bmatrix} \nabla_{\hat{\boldsymbol{\theta}}}\log p_{\hat{\boldsymbol{\theta}}}(\mathbf{y}_1|\mathbf{x}_1) \\ \vdots \\ \nabla_{\hat{\boldsymbol{\theta}}}\log p_{\hat{\boldsymbol{\theta}}}(\mathbf{y}_N|\mathbf{x}_N) \end{bmatrix} \in \mathbb{R}^{N\times D}. \tag{13}$$

This matrix is chosen such that the kernel of $\mathbf{J}_{\hat{\boldsymbol{\theta}}}^\top\mathbf{J}_{\hat{\boldsymbol{\theta}}}$ estimates the kernel of the Fisher–Rao metric, $\mathrm{ker}(\mathcal{F}_{\hat{\boldsymbol{\theta}}})$ (Equation 1), as is common practice in Bayesian deep learning (Miani et al., 2025). For a deeper discussion of the implications of this choice, we refer to Kunstner et al. (2019). Projecting onto the kernel of $\mathbf{J}_{\hat{\boldsymbol{\theta}}}^\top\mathbf{J}_{\hat{\boldsymbol{\theta}}}$ can then be written as a least-squares problem over $\mathbf{J}_{\hat{\boldsymbol{\theta}}}$,

$$\boldsymbol{\epsilon}_{\mathrm{ker}}^{(s)} = \mathbf{U}_{\hat{\boldsymbol{\theta}}}\mathbf{U}_{\hat{\boldsymbol{\theta}}}^\top\boldsymbol{\epsilon}^{(s)} = \arg\min_{\mathbf{u}}\left\{\|\mathbf{u}-\boldsymbol{\epsilon}^{(s)}\|^2 \text{ subject to } \mathbf{J}_{\hat{\boldsymbol{\theta}}}\mathbf{u}=\mathbf{0}\right\}. \tag{14}$$

Using Lagrange multipliers, we can rewrite the loss in Equation 14 as

$$\ell(\mathbf{u}, \boldsymbol{\lambda}) = \frac{1}{2}\|\mathbf{u}-\boldsymbol{\epsilon}^{(s)}\|_2^2 + \boldsymbol{\lambda}\mathbf{J}_{\hat{\boldsymbol{\theta}}}\mathbf{u},$$

with minima $\boldsymbol{\epsilon}_{\mathrm{ker}}^{(s)}$ and $\boldsymbol{\lambda}^*$ that satisfy

$$\begin{bmatrix} \mathbb{I} & \mathbf{J}_{\hat{\boldsymbol{\theta}}}^\top \\ \mathbf{J}_{\hat{\boldsymbol{\theta}}} & 0 \end{bmatrix}\begin{bmatrix} \boldsymbol{\epsilon}_{\mathrm{ker}}^{(s)} \\ \boldsymbol{\lambda}^* \end{bmatrix} = \begin{bmatrix} \boldsymbol{\epsilon}^{(s)} \\ 0 \end{bmatrix}; \tag{15}$$

more explicitly, the optimal solution is

$$\boldsymbol{\epsilon}_{\mathrm{ker}}^{(s)} = \boldsymbol{\epsilon}^{(s)} - \mathbf{J}_{\hat{\boldsymbol{\theta}}}^\top\boldsymbol{\lambda}^*, \quad \boldsymbol{\lambda}^* = (\mathbf{J}_{\hat{\boldsymbol{\theta}}}\mathbf{J}_{\hat{\boldsymbol{\theta}}}^\top)^{-1}\mathbf{J}_{\hat{\boldsymbol{\theta}}}\boldsymbol{\epsilon}^{(s)} \tag{16}$$

The numerical bottleneck of the projection is thus inverting the matrix $\mathbf{J}_{\hat{\boldsymbol{\theta}}}\mathbf{J}_{\hat{\boldsymbol{\theta}}}^\top$. We never invert this matrix explicitly, but instead solve the linear system $\mathbf{J}_{\hat{\boldsymbol{\theta}}}\mathbf{J}_{\hat{\boldsymbol{\theta}}}^\top\xi = \mathbf{J}_{\hat{\boldsymbol{\theta}}}\boldsymbol{\epsilon}^{(s)}$ with the conjugate gradients algorithm (Hestenes et al., 1952). The conjugate gradients algorithm is a matrix-free method, which means that we never materialize the system matrix, and instead, require only matrix-vector products and vector-matrix products, which are accessible with automatic differentiation (since the system matrix consists of stacks per-data gradients).

Importantly, conjugate-gradient solves can be numerically brittle as they rely on well-conditioned systems. This frequently leads to inaccurate projections, in the sense that $\boldsymbol{\epsilon}_{\mathrm{ker}}^{(s)}$ and $\boldsymbol{\epsilon}_{\mathrm{im}}^{(s)}$ are not orthogonal, even if they should. The solution to this problem is to either precondition the linear system or to improve the numerical robustness of the conjugate gradients method with reorthogonalization – we found the latter to perform well, thus, all experiments use conjugate gradients with full reorthogonalization (Gratton et al., 2021); see also Maddox et al. (2022).

## 3.3 Stochastic alternating projections

The projection algorithm performs most of the computation that eventually gets us samples from the approximate posterior $q(\boldsymbol{\theta})$. Unfortunately, following Equation 16 to project onto the FR kernel requires working with $\mathbf{J}_{\hat{\boldsymbol{\theta}}}\mathbf{J}_{\hat{\boldsymbol{\theta}}}^{\top}$, a $N \times N$ matrix. Recently, Miani et al. (2025) proposed a scalable mini-batched algorithm for projecting onto matrix kernels, based on von Neumann's alternating projections (1949). In our setting, this algorithm would iteratively project onto the kernel of batched FR estimates,

$$\boldsymbol{\epsilon} \mapsto \mathbf{U}_{\hat{\boldsymbol{\theta}}}^{(t)}\mathbf{U}_{\hat{\boldsymbol{\theta}}}^{(t)\top}\boldsymbol{\epsilon}, \tag{17}$$

where $\boldsymbol{\epsilon}$ is the vector to be projected and $\mathbf{U}_{\hat{\boldsymbol{\theta}}}^{(t)}\mathbf{U}_{\hat{\boldsymbol{\theta}}}^{(t)\top}$ is the kernel-projection of the $t^{\text{th}}$ batch. This process can be shown to converge onto the kernel of the complete-data FR (Miani et al., 2025).

Unfortunately, the alternating projections algorithm still requires one or more passes through the entire dataset to project a single vector. As we require a sample from the approximate posterior in every optimization step, alternating projections is inapplicable to mini-batched ELBO optimization.

The fundamental issue is that in each ELBO optimization step, we change the approximate posterior mean $\hat{\boldsymbol{\theta}}$, such that the FR and its kernel also change. A naive procedure would draw an initial sample $\boldsymbol{\epsilon}^{(0)} \sim \mathcal{N}(\mathbf{0}, \mathbb{I})$, and repeatedly project this onto mini-batch FR kernels throughout ELBO optimization, hoping that the posterior mean changes sufficiently slowly to maintain the properties of kernel samples. Empirically, we have not found this to be the case.

With this in mind, we propose a stochastic extension of the alternating projections algorithm suitable for mini-batched ELBO optimization. Let $\boldsymbol{\epsilon}^{(t)}$ denote a sample in the FR kernel associated with the posterior mean at step $t$ (e.g., one parameter update step). We then update the sample as

$$\boldsymbol{\epsilon}^{(t)} = \mathbf{U}_{\hat{\boldsymbol{\theta}}}^{(t)}\mathbf{U}_{\hat{\boldsymbol{\theta}}}^{(t)\top}\left(\sqrt{\gamma}\,\boldsymbol{\epsilon}^{(t-1)} + \sqrt{1-\gamma}\,\boldsymbol{\eta}^{(t)}\right), \qquad \boldsymbol{\eta}^{(t)} \sim \mathcal{N}(0, \mathbb{I}), \tag{18}$$

where the hyperparameter $\gamma \in [0, 1]$ controls the noise of this stochastic alternating projection. For $\gamma = 1$, we recover the above naive approach that disregards the change in kernel, while for $\gamma = 0$ we merely project onto the kernel of the current batched FR estimate. In-between values implement a soft "sliding window" along the history of projections, where previous projection information is preserved and then eventually forgotten after a certain number of steps.

**What is the effect of additional noise?** Empirically, we find the stochastic extension essential to training models successfully. In Figure 2, we show the accuracy curves over training of a small CNN classifier on Fashion MNIST, with dashed horizontal lines indicating performance on test data. The experiment evaluates values of $\gamma$: no noise ($\gamma = 1.0$), only noise ($\gamma = 0.0$), and an in-between choice ($\gamma = 0.5$), as per Equation 18. The model without noise quickly achieves the highest training accuracy, but its dependency on the projected samples, which change only once per epoch, leads to a poorer approximate posterior and generalization.

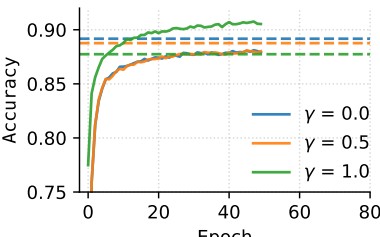

Figure 2: Performance of $\gamma$ values.

## 3.4 Post-hoc variational inference or full model fitting?

Given the computational effort to evaluate and run the ELBO, it is important to question whether this could be avoided entirely. One possibility is to start from a well-performing model, and only use the ELBO to tune the variational parameters $\sigma_{\text{ker}}$ and $\sigma_{\text{im}}$, i.e., post-hoc variational inference with a fixed model. This can be efficiently implemented as we only need to project one set of samples, which can be reused throughout optimization. With the same experimental setting from before, Figure 3 shows the behavior of such a training procedure. Interestingly, post-hoc tuning is not *per se* prob-

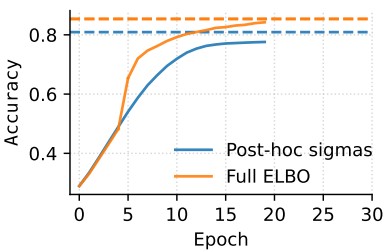

Figure 3: Post-hoc tuning $\sigma_{\text{ker}}, \sigma_{\text{im}}$.

lematic and indeed leads to a well-performing model. However, the figure shows that optimizing the variational mean with the ELBO improves upon the post-hoc approach.

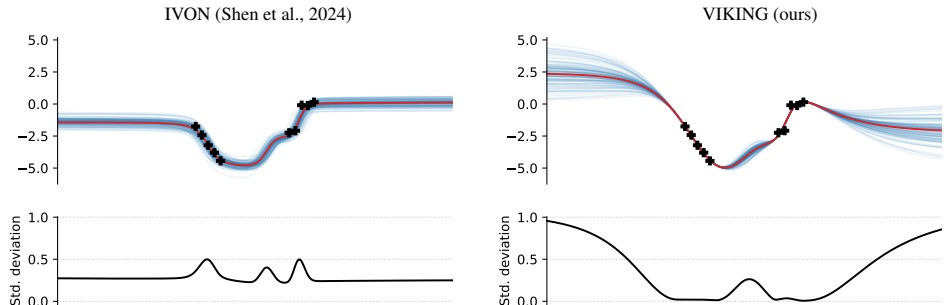

Figure 5: A toy regression example on a sinusoid curve with 10 data points. *Top:* The curves show the training points in black, with the mean fit as a red line and 100 posterior predictive samples as blue lines. *Bottom:* The standard deviation of the predictions over each point in the horizontal axis.

**Warming up as a shortcut.** As observed, we can still obtain a well-behaved model from the post-hoc tuning procedure. Naturally, this informally suggests another way to save computational effort, resources, and time: "warming up" the model, first with regular maximum likelihood estimation (or through a pretrained model), and then continuing with ELBO optimization.

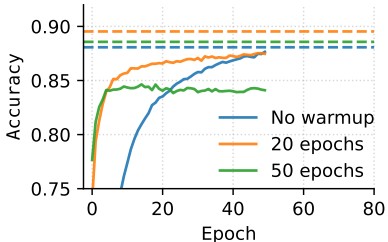

Figure 4: Warmup by pretraining.

We test this hypothesis experimentally, comparing a model trained from scratch using our ELBO and one that starts with a model first fitted using maximum likelihood (Figure 4). Our findings are twofold: (i) warmup using maximum likelihood is an effective way to quickstart the ELBO learning process with the ELBO; (ii) there is a sweet spot where full convergence of a model on maximum likelihood can lead to the ELBO optimization getting stuck. We conjecture that this behavior is due to the sharpness of some minima, which negatively affects the evaluation of the expectation term, as it attempts to sample regions of the weight space around the current model weights, which could have drastically different performance, negatively affecting optimization.

## 4 Experiments

We evaluate VIKING against popular Bayesian deep learning methods using standard benchmarks for Bayesian deep learning. Aiming to maintain consistency and simplify comparisons, we adopt the loss-Jacobian (see Section 3.2) when using our proposed variational family. Further details on models, hyperparameters used, and training procedures are in Section C.

**Toy (sinusoid) regression.** We first illustrate posterior samples in a 1D regression in Figure 5. We train the same model with both IVON (Shen et al., 2024) and VIKING. While IVON obtains a good model fit, its posterior samples are uninformative with respect to uncertainties. On the other hand, VIKING shows higher variance close to the boundary of data and beyond, where the model should be less confident.

**Image classification.** Using standard benchmarks, we evaluate our method against the *maximum a posteriori* (MAP) estimate, the (loss-projected) post-hoc method from Miani et al. (2025), IVON (Shen et al., 2024), SWAG (Maddox et al., 2019), and a last-layer Laplace approximation. On MNIST (LeCun et al., 2010) and Fashion MNIST (Xiao et al., 2017), we train a LeNet (LeCun et al., 1989) model, while on SVHN (Netzer et al., 2011) and CIFAR-10 (Krizhevsky and Hinton, 2009) we use a small ResNet (He et al., 2016). The models are compared using accuracy, negative log-likelihood (NLL), expected calibration error (ECE), and maximum calibration error (MCE) (Naeini et al., 2015).

The results in Table 1 show that our ELBO generalize better in most cases, which we hypothesize is due to the interaction between evaluations of the expectation term which constantly "explores" the weight space around the current mode through the sampling mechanism, whereas the kernel and image variances that allow this exploration are kept in check by the KL term. On calibration metrics, our method is particularly effective against the baselines on the SVHN and CIFAR-10 datasets, where

Table 1: Experimental results over three runs (mean and standard deviation) on in-distribution test data. MAP is a point (model) estimate. VIKING and the post-hoc method from Miani et al. (2025) use the loss-projected variant.

| | | Accuracy$^\uparrow$ | Conf.$^\uparrow$ | NLL$^\downarrow$ | ECE$^\downarrow$ | MCE$^\downarrow$ |
|---|---|---|---|---|---|---|
| MNIST | MAP | $0.986 \pm 0.001$ | $\mathbf{0.996} \pm \mathbf{0.000}$ | $0.070 \pm 0.005$ | $0.247 \pm 0.011$ | $0.861 \pm 0.045$ |
| | VIKING (ours) | $\mathbf{0.991} \pm \mathbf{0.001}$ | $0.992 \pm 0.000$ | $0.055 \pm 0.003$ | $0.096 \pm 0.004$ | $0.690 \pm 0.102$ |
| | Miani et al. (2025) | $0.949 \pm 0.000$ | $0.813 \pm 0.018$ | $1.225 \pm 0.099$ | $0.666 \pm 0.007$ | $0.894 \pm 0.011$ |
| | IVON | $0.989 \pm 0.001$ | $0.990 \pm 0.001$ | $\mathbf{0.043} \pm \mathbf{0.002}$ | $\mathbf{0.077} \pm \mathbf{0.005}$ | $\mathbf{0.651} \pm \mathbf{0.042}$ |
| | SWAG | $0.982 \pm 0.000$ | $0.982 \pm 0.001$ | $0.064 \pm 0.006$ | $0.788 \pm 0.005$ | $0.906 \pm 0.013$ |
| | Last Layer LA | $0.975 \pm 0.002$ | $0.977 \pm 0.002$ | $0.090 \pm 0.005$ | $0.784 \pm 0.007$ | $0.887 \pm 0.008$ |
| Fashion MNIST | MAP | $0.883 \pm 0.002$ | $\mathbf{0.942} \pm \mathbf{0.003}$ | $0.410 \pm 0.010$ | $0.153 \pm 0.008$ | $\mathbf{0.590} \pm \mathbf{0.141}$ |
| | VIKING (ours) | $\mathbf{0.900} \pm \mathbf{0.001}$ | $0.928 \pm 0.001$ | $0.332 \pm 0.003$ | $0.075 \pm 0.002$ | $0.611 \pm 0.160$ |
| | Miani et al. (2025) | $0.871 \pm 0.006$ | $0.744 \pm 0.031$ | $1.529 \pm 0.371$ | $0.617 \pm 0.025$ | $0.901 \pm 0.013$ |
| | IVON | $0.897 \pm 0.004$ | $0.926 \pm 0.001$ | $0.335 \pm 0.011$ | $\mathbf{0.073} \pm \mathbf{0.005}$ | $0.683 \pm 0.024$ |
| | SWAG | $0.898 \pm 0.001$ | $0.931 \pm 0.006$ | $\mathbf{0.327} \pm \mathbf{0.001}$ | $0.725 \pm 0.003$ | $0.907 \pm 0.003$ |
| | Last Layer LA | $0.896 \pm 0.002$ | $0.931 \pm 0.005$ | $0.339 \pm 0.011$ | $0.727 \pm 0.004$ | $0.902 \pm 0.004$ |
| SVHN | MAP | $0.947 \pm 0.004$ | $0.963 \pm 0.004$ | $0.201 \pm 0.014$ | $0.055 \pm 0.010$ | $0.608 \pm 0.228$ |
| | VIKING (ours) | $\mathbf{0.960} \pm \mathbf{0.001}$ | $\mathbf{0.964} \pm \mathbf{0.001}$ | $\mathbf{0.177} \pm \mathbf{0.002}$ | $\mathbf{0.028} \pm \mathbf{0.002}$ | $\mathbf{0.308} \pm \mathbf{0.024}$ |
| | Miani et al. (2025) | $0.949 \pm 0.003$ | $0.948 \pm 0.005$ | $0.191 \pm 0.007$ | $0.734 \pm 0.017$ | $0.880 \pm 0.012$ |
| | IVON | $0.943 \pm 0.002$ | $0.888 \pm 0.007$ | $0.302 \pm 0.016$ | $0.082 \pm 0.004$ | $0.492 \pm 0.248$ |
| | SWAG | $0.947 \pm 0.004$ | $0.897 \pm 0.007$ | $0.217 \pm 0.014$ | $0.745 \pm 0.007$ | $0.874 \pm 0.003$ |
| | Last Layer LA | $0.946 \pm 0.001$ | $0.943 \pm 0.005$ | $0.197 \pm 0.009$ | $0.740 \pm 0.007$ | $0.899 \pm 0.009$ |
| CIFAR-10 | MAP | $0.824 \pm 0.012$ | $0.869 \pm 0.003$ | $0.536 \pm 0.055$ | $0.075 \pm 0.012$ | $0.619 \pm 0.243$ |
| | VIKING (ours) | $0.877 \pm 0.004$ | $0.893 \pm 0.003$ | $0.407 \pm 0.010$ | $\mathbf{0.041} \pm \mathbf{0.004}$ | $\mathbf{0.331} \pm \mathbf{0.094}$ |
| | Miani et al. (2025) | $0.855 \pm 0.002$ | $0.701 \pm 0.013$ | $2.643 \pm 0.205$ | $0.559 \pm 0.006$ | $0.802 \pm 0.005$ |
| | IVON | $0.835 \pm 0.017$ | $0.763 \pm 0.005$ | $0.817 \pm 0.075$ | $0.086 \pm 0.014$ | $0.436 \pm 0.244$ |
| | SWAG | $0.865 \pm 0.029$ | $0.914 \pm 0.035$ | $0.445 \pm 0.063$ | $0.694 \pm 0.018$ | $0.881 \pm 0.005$ |
| | Last Layer LA | $\mathbf{0.894} \pm \mathbf{0.001}$ | $\mathbf{0.944} \pm \mathbf{0.001}$ | $\mathbf{0.406} \pm \mathbf{0.005}$ | $0.704 \pm 0.000$ | $0.880 \pm 0.007$ |
| Imagenette | MAP | $0.852 \pm 0.002$ | $0.883 \pm 0.007$ | $0.481 \pm 0.009$ | $0.084 \pm 0.010$ | $0.717 \pm 0.082$ |
| | VIKING (ours) | $\mathbf{0.887} \pm \mathbf{0.003}$ | $\mathbf{0.906} \pm \mathbf{0.002}$ | $\mathbf{0.403} \pm \mathbf{0.010}$ | $0.077 \pm 0.001$ | $0.612 \pm 0.162$ |
| | IVON | $0.876 \pm 0.023$ | $0.849 \pm 0.013$ | $0.656 \pm 0.136$ | $\mathbf{0.069} \pm \mathbf{0.011}$ | $\mathbf{0.464} \pm \mathbf{0.230}$ |

the overparametrization of the model (around 220k parameters) is more prominent. Additional results including deep ensembles and SGLD (Welling and Teh, 2011) are in Section A.

**Does it scale to larger models?**  To evaluate the scalability of our approach, we also train a ResNet34 (21.7 million parameters) on Imagenette (Howard, 2019) using VIKING and IVON. We predominantly aim to demonstrate that VIKING scales to contemporary models despite learning a fully correlated covariance. The results show a superior performance of our ELBO-trained model, but inferior calibration metrics compared to IVON (Table 1).

**Out-of-distribution detection.**  Next, we evaluate VIKING in out-of-distribution (OOD) detection tasks. We use the maximum variance of the softmax probabilities across output dimensions as an OOD score for both IVON and VIKING. In addition to the datasets from before, we use EMNIST and CIFAR-100 as OOD data. For ease of comparison, our experimental setup replicates that of Miani et al. (2025), so we additionally use their reported numbers for the other baselines as well.

Table 2 summarizes the obtained results. At a glance, VIKING performs on par with the best models, sometimes overperforming the other baselines by a big margin (e.g., MNIST $\to$ FMNIST and KMNIST) and other times being a close second. Overall, this further confirms the empirical performance of our variational family and ELBO optimization in obtaining robust models.

**Generative modelling.**  Finally, we demonstrate our approach applied to generative modelling. We train a variational autoencoder (VAE, Kingma and Welling (2013)) until convergence. From that starting point, we further refine the decoder's 1.6 million parameters using both IVON and VIKING. We draw 32 samples from each of the resulting approximate decoder posteriors, which we use to both reconstruct images and measure uncertainties.

Figure 6 displays the reconstructed samples and the variance across the posterior samples per reconstructed image. Overall, both posteriors qualitatively recover well-performing models. The methods discern themselves more prominently upon inspection of the variances across the reconstructions of

Table 2: Area under ROC (↑) performance on out-of-distribution detection. VIKING and the post-hoc method from Miani et al. (2025) use the loss-projected variant.

| In-dist.
Out-of-dist. | MNIST
FMNIST | 
KMNIST | 
EMNIST | FMNIST
MNIST | 
KMNIST | 
EMNIST |
|---|---|---|---|---|---|---|
| MAP | $0.928 \pm 0.015$ | $0.934 \pm 0.004$ | $0.890 \pm 0.002$ | $0.728 \pm 0.048$ | $0.791 \pm 0.014$ | $0.659 \pm 0.035$ |
| VIKING (ours) | $\mathbf{0.972} \pm \mathbf{0.003}$ | $\mathbf{0.965} \pm \mathbf{0.002}$ | $0.913 \pm 0.006$ | $0.898 \pm 0.025$ | $\mathbf{0.926} \pm \mathbf{0.002}$ | $0.830 \pm 0.020$ |
| Miani et al. (2025) | $0.899 \pm 0.011$ | $0.856 \pm 0.002$ | $0.893 \pm 0.006$ | $\mathbf{0.914} \pm \mathbf{0.035}$ | $0.907 \pm 0.026$ | $\mathbf{0.928} \pm \mathbf{0.007}$ |
| IVON | $0.948 \pm 0.004$ | $0.945 \pm 0.007$ | $\mathbf{0.918} \pm \mathbf{0.008}$ | $0.801 \pm 0.020$ | $0.867 \pm 0.005$ | $0.739 \pm 0.025$ |
| SWAG | $0.917 \pm 0.024$ | $0.861 \pm 0.032$ | $0.916 \pm 0.010$ | $0.685 \pm 0.014$ | $0.655 \pm 0.015$ | $0.751 \pm 0.032$ |
| Last Layer LA | $0.793 \pm 0.215$ | $0.759 \pm 0.166$ | $0.782 \pm 0.182$ | $0.768 \pm 0.001$ | $0.699 \pm 0.020$ | $0.824 \pm 0.016$ |

| In-dist.
Out-of-dist. | SVHN
CIFAR-10 | 
CIFAR-100 | CIFAR-10
SVHN | 
CIFAR-100 |
|---|---|---|---|---|
| MAP | $0.946 \pm 0.006$ | $0.941 \pm 0.007$ | $0.831 \pm 0.019$ | $0.778 \pm 0.013$ |
| VIKING (ours) | $0.947 \pm 0.002$ | $0.943 \pm 0.002$ | $0.827 \pm 0.007$ | $\mathbf{0.805} \pm \mathbf{0.003}$ |
| Miani et al. (2025) | $\mathbf{0.966} \pm \mathbf{0.009}$ | $\mathbf{0.960} \pm \mathbf{0.006}$ | $\mathbf{0.863} \pm \mathbf{0.008}$ | $0.800 \pm 0.013$ |
| IVON | $0.822 \pm 0.014$ | $0.826 \pm 0.009$ | $0.648 \pm 0.019$ | $0.714 \pm 0.012$ |
| SWAG | $0.777 \pm 0.029$ | $0.787 \pm 0.027$ | $0.798 \pm 0.040$ | $0.782 \pm 0.042$ |
| Last Layer LA | $0.914 \pm 0.005$ | $0.908 \pm 0.005$ | $0.811 \pm 0.024$ | $0.801 \pm 0.026$ |

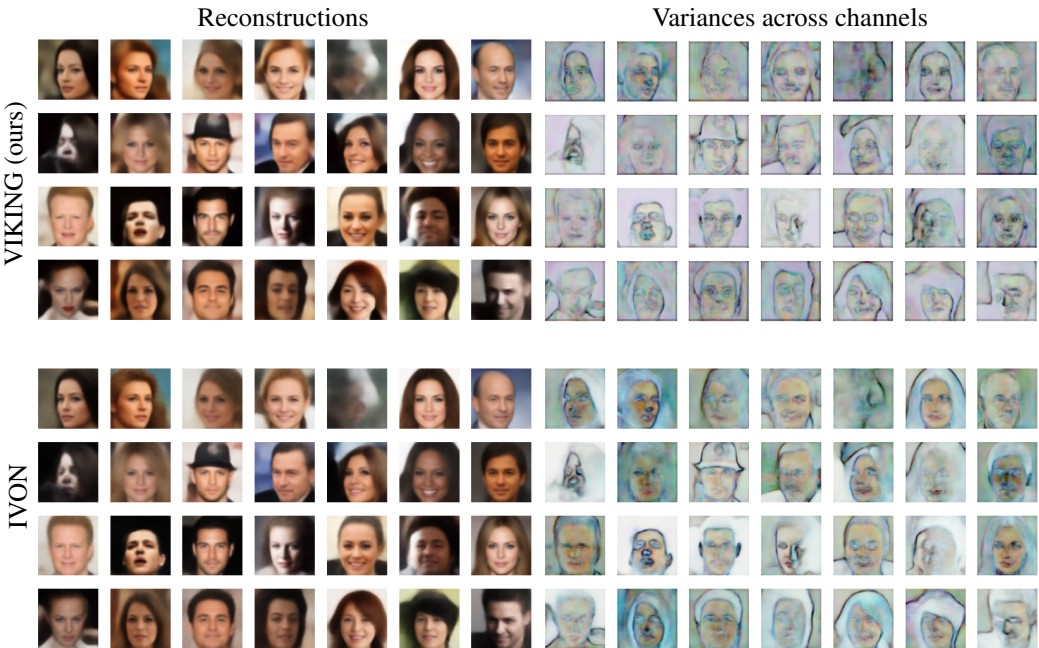

Figure 6: Reconstructed samples taken from the latent space prior of variational autoencoders trained with IVON and VIKING. Using 32 posterior samples, the posterior variances compare key semantic features, such as contours between foreground and background, and eyes.

the different posterior samples. Notably, IVON tends to capture variance across all features of the image, including the background. VIKING focuses the variance on facial features and outline instead, showing a disregard for the less-relevant aspects of the data.

To assess the model uncertainties, we compute the median value of the standard deviations per pixel produced by each of the 16k generated samples and summarize their distribution in Figure 7. Our in-distribution samples come from a standard Gaussian sample, whereas out-of-distribution samples come from a Gaussian with twice the variance. The uncertainties produced by VIKING show well-separated distributions clearly detecting OOD samples, while IVON does not.

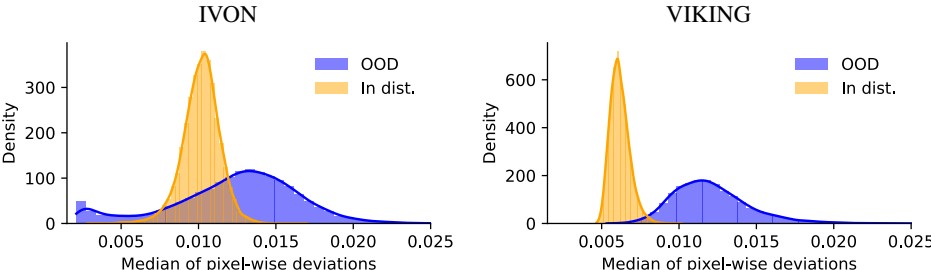

Figure 7: Histograms of uncertainties on 16K in- and out-of-distribution latent samples according to IVON and VIKING. Uncertainties are computed as median of pixel-wise standard deviations computed by decoding the latent sample using 32 posterior model samples.

## 5 Discussion

This paper introduces a variational family ('VIKING') that explicitly takes model overparametrization into account. This is achieved through a decomposition of the Fisher–Rao metric into parameter subspaces that capture directions in which the per-training data loss remains unchanged and directions in which this loss changes. Intuitively, uncertainty along these directions can be interpreted as uncertainty over the training data and general model uncertainty. This is similar to the conventional splits into aleatoric and epistemic uncertainty commonly found, e.g., in Gaussian processes (GPs, Rasmussen and Williams 2006). The parameters we estimate, thus, come with a degree of interpretability.

Scaling VIKING to large models is, however, not straightforward, as the approximate posterior correlates all parameters, yet we cannot afford to instantiate the covariance matrix due to excessive memory costs. We have developed a stochastic extension to von Neumann's (1949) alternating projections algorithm that scales to large models through a matrix-free implementation. The approach readily applies to large contemporary models including a ResNet with 21.7 million parameters and a generative model with 1.6 million parameters.

Empirically, our results show that VIKING tends to be *consistently* as good as or better than current state-of-the-art methods. This lends credibility to the hypothesis that overparametrization should be explicitly taken into account when designing approximate posteriors.

**The key limitation** is the additional computational overhead incurred by the projection algorithm. Standard gradient descent requires a single backward pass for one gradient update, while VIKING requires the number of CG iterations backward passes per sample. In practice, we train with one Monte Carlo sample and only need a few CG iterations, so the additional cost is manageable. Using more sophisticated linear algebra and custom gradients, significantly more efficient and numerically accurate least squares solvers can be employed (Roy et al., 2025), overcoming some of the time-efficiency and numerical issues in our approach.

## Acknowledgments

The authors thank Sebastian Mair for his comments on early drafts. This work was supported by the Danish Data Science Academy, which is funded by the Novo Nordisk Foundation (NNF21SA0069429); by research grant 42062 from VILLUM FONDEN; by the Novo Nordisk Foundation through the *Center for Basic Machine Learning Research in Life Science* (NNF20OC0062606); by the European Research Council (ERC) under the European Union's Horizon programme (grant agreement 101125993); and by the Horizon-EU EIC Pathfinder Open program to *i-RASE*: Intelligent Radiation Sensor Readout System, with reference number: *HE EIC - i-RASE – 101130550*.

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

Table 3: Experimental results over three runs (mean and standard deviation) on in-distribution test data. MAP is a point (model) estimate. VIKING and the post-hoc method from Miani et al. (2025) use the loss-projected variant. Ensemble is over five models.

| | | Accuracy$^\uparrow$ | Conf.$^\uparrow$ | NLL$^\downarrow$ | ECE$^\downarrow$ | MCE$^\downarrow$ |
|---|---|---|---|---|---|---|
| MNIST | MAP | $0.986 \pm 0.001$ | $\mathbf{0.996} \pm \mathbf{0.000}$ | $0.070 \pm 0.005$ | $0.247 \pm 0.011$ | $0.861 \pm 0.045$ |
| | VIKING (ours) | $0.991 \pm 0.001$ | $0.992 \pm 0.000$ | $0.055 \pm 0.003$ | $0.096 \pm 0.004$ | $0.690 \pm 0.102$ |
| | Miani et al. (2025) | $0.949 \pm 0.000$ | $0.813 \pm 0.018$ | $1.225 \pm 0.099$ | $0.666 \pm 0.007$ | $0.894 \pm 0.011$ |
| | IVON | $0.989 \pm 0.001$ | $0.990 \pm 0.001$ | $\mathbf{0.043} \pm \mathbf{0.002}$ | $0.077 \pm 0.005$ | $\mathbf{0.651} \pm \mathbf{0.042}$ |
| | SGLD | $0.990 \pm 0.000$ | $0.975 \pm 0.001$ | $0.062 \pm 0.002$ | $\mathbf{0.067} \pm \mathbf{0.002}$ | $0.684 \pm 0.054$ |
| | SWAG | $0.982 \pm 0.000$ | $0.982 \pm 0.001$ | $0.064 \pm 0.006$ | $0.788 \pm 0.005$ | $0.906 \pm 0.013$ |
| | Last Layer LA | $0.975 \pm 0.002$ | $0.977 \pm 0.002$ | $0.090 \pm 0.005$ | $0.784 \pm 0.007$ | $0.887 \pm 0.008$ |
| | Ensemble | $\mathbf{0.992}$ | $0.989$ | $0.078$ | $0.116$ | $0.675$ |
| Fashion MNIST | MAP | $0.883 \pm 0.002$ | $\mathbf{0.942} \pm \mathbf{0.003}$ | $0.410 \pm 0.010$ | $0.153 \pm 0.008$ | $\mathbf{0.590} \pm \mathbf{0.141}$ |
| | VIKING (ours) | $0.900 \pm 0.001$ | $0.928 \pm 0.001$ | $0.332 \pm 0.003$ | $0.075 \pm 0.002$ | $0.611 \pm 0.160$ |
| | Miani et al. (2025) | $0.871 \pm 0.006$ | $0.744 \pm 0.031$ | $1.529 \pm 0.371$ | $0.617 \pm 0.025$ | $0.901 \pm 0.013$ |
| | IVON | $0.897 \pm 0.004$ | $0.926 \pm 0.001$ | $0.335 \pm 0.011$ | $0.073 \pm 0.005$ | $0.683 \pm 0.024$ |
| | SGLD | $0.899 \pm 0.001$ | $0.899 \pm 0.001$ | $\mathbf{0.316} \pm \mathbf{0.003}$ | $\mathbf{0.026} \pm \mathbf{0.004}$ | $0.734 \pm 0.019$ |
| | SWAG | $0.898 \pm 0.001$ | $0.931 \pm 0.006$ | $0.327 \pm 0.001$ | $0.725 \pm 0.003$ | $0.907 \pm 0.003$ |
| | Last Layer LA | $0.896 \pm 0.002$ | $0.931 \pm 0.005$ | $0.339 \pm 0.011$ | $0.727 \pm 0.004$ | $0.902 \pm 0.004$ |
| | Ensemble | $\mathbf{0.910}$ | $0.913$ | $0.670$ | $0.052$ | $0.793$ |
| SVHN | MAP | $0.947 \pm 0.004$ | $0.963 \pm 0.004$ | $0.201 \pm 0.014$ | $0.055 \pm 0.010$ | $0.608 \pm 0.228$ |
| | VIKING (ours) | $0.960 \pm 0.001$ | $\mathbf{0.964} \pm \mathbf{0.001}$ | $\mathbf{0.177} \pm \mathbf{0.002}$ | $\mathbf{0.028} \pm \mathbf{0.002}$ | $\mathbf{0.308} \pm \mathbf{0.024}$ |
| | Miani et al. (2025) | $0.949 \pm 0.003$ | $0.948 \pm 0.005$ | $0.191 \pm 0.007$ | $0.734 \pm 0.017$ | $0.880 \pm 0.012$ |
| | IVON | $0.943 \pm 0.002$ | $0.888 \pm 0.007$ | $0.302 \pm 0.016$ | $0.082 \pm 0.004$ | $0.492 \pm 0.248$ |
| | SGLD | $0.698 \pm 0.103$ | $0.378 \pm 0.027$ | $1.935 \pm 0.070$ | $0.321 \pm 0.128$ | $0.605 \pm 0.181$ |
| | SWAG | $0.947 \pm 0.004$ | $0.897 \pm 0.007$ | $0.217 \pm 0.014$ | $0.745 \pm 0.007$ | $0.874 \pm 0.003$ |
| | Last Layer LA | $0.946 \pm 0.001$ | $0.943 \pm 0.005$ | $0.197 \pm 0.009$ | $0.740 \pm 0.007$ | $0.899 \pm 0.009$ |
| | Ensemble | $\mathbf{0.965}$ | $0.955$ | $0.231$ | $0.051$ | $0.769$ |
| CIFAR-10 | MAP | $0.824 \pm 0.012$ | $0.869 \pm 0.003$ | $0.536 \pm 0.055$ | $0.075 \pm 0.012$ | $0.619 \pm 0.243$ |
| | VIKING (ours) | $0.877 \pm 0.004$ | $0.893 \pm 0.003$ | $0.407 \pm 0.010$ | $\mathbf{0.041} \pm \mathbf{0.004}$ | $\mathbf{0.331} \pm \mathbf{0.094}$ |
| | Miani et al. (2025) | $0.855 \pm 0.002$ | $0.701 \pm 0.013$ | $2.643 \pm 0.205$ | $0.559 \pm 0.006$ | $0.802 \pm 0.005$ |
| | IVON | $0.835 \pm 0.017$ | $0.763 \pm 0.005$ | $0.817 \pm 0.075$ | $0.086 \pm 0.014$ | $0.436 \pm 0.244$ |
| | SGLD | $0.495 \pm 0.030$ | $0.345 \pm 0.020$ | $2.409 \pm 0.135$ | $0.153 \pm 0.039$ | $0.375 \pm 0.073$ |
| | SWAG | $0.865 \pm 0.029$ | $0.914 \pm 0.035$ | $0.445 \pm 0.063$ | $0.694 \pm 0.018$ | $0.881 \pm 0.005$ |
| | Last Layer LA | $0.894 \pm 0.001$ | $\mathbf{0.944} \pm \mathbf{0.001}$ | $\mathbf{0.406} \pm \mathbf{0.005}$ | $0.704 \pm 0.000$ | $0.880 \pm 0.007$ |
| | Ensemble | $\mathbf{0.919}$ | $0.891$ | $0.486$ | $0.064$ | $0.780$ |
| Imagenette | MAP | $0.852 \pm 0.002$ | $0.883 \pm 0.007$ | $0.481 \pm 0.009$ | $0.084 \pm 0.010$ | $0.717 \pm 0.082$ |
| | VIKING (ours) | $\mathbf{0.887} \pm \mathbf{0.003}$ | $\mathbf{0.906} \pm \mathbf{0.002}$ | $\mathbf{0.403} \pm \mathbf{0.010}$ | $0.077 \pm 0.001$ | $0.612 \pm 0.162$ |
| | IVON | $0.876 \pm 0.023$ | $0.849 \pm 0.013$ | $0.656 \pm 0.136$ | $\mathbf{0.069} \pm \mathbf{0.011}$ | $\mathbf{0.464} \pm \mathbf{0.230}$ |

# A  Additional experimental results

In Table 3 we detail the same experiments from the main paper, but with two additional baselines. The first is a simple deep ensemble, where five copies of the model are trained independently and their predictions are averaged. Here, the best-performing MAP model hyperparameters from the existing experiments are used to train the models again using additional seeds to build the ensemble.

The second is Stochastic Gradient Langevin Dynamics (SGLD, Welling and Teh, 2011), which is implemented in JAX using code provided by Izmailov et al. (2021). The hyperparameters are optimized using a grid search with a similar budget we use for VIKING, using experimental evidence by Izmailov et al. (2021) as a guideline. When collecting posterior samples from SGLD, we allowed for a burn-in period of 50 epochs, then collected each posterior sample after additional 50 epochs are done, totalling 300 epochs, which approximately corresponds to 50 epochs of ELBO training with VIKING.

These baselines are included as additional evidence that, given similar compute budgets, VIKING outperforms classical approaches such as SGLD and deep ensembles in calibration metrics, while still achieving competitive performance. Notably, deep ensembles achieve the best predictive performance, but that is not reflected in the calibration metrics. SGLD, on the other hand, struggled in the harder datasets with bigger models (SVHN, CIFAR-10), showing that the compute budget available is probably not enough, needing either a significantly larger number of training steps to converge or hyperparameter tuning. When it does properly converge, such as on Fashion MNIST and MNIST, its performance is not far behind the state of the art.

## B    Additional methodological details

Here we expand on the rationale and discussion present in the main paper. Our aim is to address potential questions, issues, and minor challenges pertaining the formulation and use of our variational family. For notational simplicity, we omit the dependence on $\boldsymbol{\theta}$ in the analysis below where convenient.

### B.1    Is the rank $R$ differentiable?

We know that the derivative of the rank $R$ with respect to $\hat{\boldsymbol{\theta}}$ is either zero or undefined, as $R$ is integer-valued and bound to be piecewise-constant. Its importance in optimizing the KL (Kullback-Leibler divergence) term is clearer by first assuming that $\partial R/\partial \hat{\theta} = 0$. Then,

$$
\begin{aligned}
\frac{\partial \mathrm{KL}}{\partial \hat{\boldsymbol{\theta}}} &= \frac{\partial}{\partial \hat{\boldsymbol{\theta}}} \left\{ \frac{1}{2} \left( \alpha \mathrm{Tr}(\boldsymbol{\Sigma}) - D + \alpha \|\hat{\boldsymbol{\theta}}\|^2 - D \log(\alpha) - \log \det(\boldsymbol{\Sigma}) \right) \right\} \\
&= \frac{\partial}{\partial \hat{\boldsymbol{\theta}}} \left\{ \frac{1}{2} \left( \alpha(\sigma_{\mathrm{ker}}^2 R + \sigma_{\mathrm{im}}^2(D-R)) + \alpha \|\hat{\boldsymbol{\theta}}\|^2 - 2R \log(\sigma_{\mathrm{ker}}) - 2(D-R) \log(\sigma_{\mathrm{im}}) \right) \right\} \\
&= \frac{\partial}{\partial \hat{\boldsymbol{\theta}}} \left\{ \frac{\alpha}{2} \|\hat{\boldsymbol{\theta}}\|^2 \right\}.
\end{aligned}
\tag{19}
$$

Under this assumption, the KL term reduces to a more usual $\ell_2$ regularization. In practice, since the estimated $R$ is not an integer, this assumption does not become problematic. In our implementation, we stop the gradients of the computed $R$ to ensure this.

### B.2    Understanding the KL term in isolation

Let us consider the kernel component of the posterior in isolation and the KL term of the ELBO $\mathcal{L}$ (Equation 3) separately to interpret what signal might exist in that term and why that can influence our hyperparameter choices.

Suppose our posterior covariance is given by $\boldsymbol{\Sigma}_q \in \mathbb{R}^{D \times D}$ and our prior covariance is given by $\boldsymbol{\Sigma}_p \in \mathbb{R}^{D \times D}$. Due to the restrictions to the subspace, we can write $\boldsymbol{\Sigma}_q = \mathbf{U}\boldsymbol{\Lambda}\mathbf{U}^\top$ and $\boldsymbol{\Sigma}_p = \alpha^{-1}\mathbb{I}$. We want to find the $\boldsymbol{\Lambda}$ that minimizes the KL term. We can write the terms that depend only on $\boldsymbol{\Sigma}_q$ as

$$
\mathrm{KL} = \mathrm{Tr}(\boldsymbol{\Sigma}_p^{-1}\boldsymbol{\Sigma}_q) - \log |\boldsymbol{\Sigma}_p^{-1}\boldsymbol{\Sigma}_q|.
$$

Note that

$$
\begin{aligned}
\boldsymbol{\Sigma}_p^{-1}\boldsymbol{\Sigma}_q &= \left( \alpha^{-1}\mathbb{I} \right)^{-1} \mathbf{U}\boldsymbol{\Lambda}\mathbf{U}^\top \\
&= \mathbf{U}(\alpha\boldsymbol{\Lambda})\mathbf{U}^\top,
\end{aligned}
$$

implying the eigenvalues of $\boldsymbol{\Sigma}_p^{-1}\boldsymbol{\Sigma}_q$ are given by $\alpha\boldsymbol{\Lambda}$, yielding

$$
\mathrm{Tr}(\mathbf{U}(\alpha\boldsymbol{\Lambda})\mathbf{U}^\top) = \sum_{i=1}^D \alpha\lambda_i, \quad \text{and} \quad \log |\mathbf{U}(\alpha\boldsymbol{\Lambda})\mathbf{U}^\top| = \sum_{i=1}^D \log(\alpha\lambda_i).
$$

For all $i = 1, \ldots, D$, $\lambda_i = \alpha^{-1}$ minimizes $f(\alpha\lambda_i) = \alpha\lambda_i - \log(\alpha\lambda_i)$ . The covariance that minimizes the KL divergence is thus

$$
\boldsymbol{\Sigma} = \alpha^{-1}\mathbf{U}\mathbf{U}^\top.
\tag{20}
$$

Since our posterior is $\boldsymbol{\Sigma}_q := \sigma_{\mathrm{ker}}^2 \mathbf{U}\mathbf{U}^\top + \sigma_{\mathrm{im}}^2(\mathbb{I} - \mathbf{U}\mathbf{U}^\top)$, we optimize $\alpha$ and $\sigma_{\mathrm{ker}}^2$ jointly by always ensuring $\alpha^{-1} = \sigma_{\mathrm{ker}}^2 \Leftrightarrow \alpha = 1/\sigma_{\mathrm{ker}}^2$.

## C    Experimental details

For reproduction purposes, we detail each experiment (in order of appearance in the main paper). Note that the code implementing VIKING and for reproducing the experiments is publicly available[4].

---

[4]`https://github.com/eugene/viking-paper-experiments`

We start with a few general experimental and implementation details used across the experiments. All optimization steps were done using the Adam optimizer. In what follows, we address specific components of the optimization that are shared across all experiments, unless otherwise noted.

**Variational parameters $\sigma_{\mathrm{ker}}$ and $\sigma_{\mathrm{im}}$.**  In all experiments, the variational parameter $\sigma_{\mathrm{ker}}$ is tied to the prior precision $\alpha$, such that $\alpha = 1/\sigma_{\mathrm{ker}}^2$ is always true (for the reasoning, see Equation 20, in Section B) and use $\log \alpha$ as an optimization variable. Furthermore, we observe that the posterior samples used to evaluate the expectation term of the ELBO can be quite sensitive to larger values of $\sigma_{\mathrm{im}}$, especially since only the kernel part of the projection is guaranteed to not deviate arbitrarily from the MAP predictions. For this reason, we always initialize this variational parameter such that $\log \sigma_{\mathrm{im}} = -2$ to prevent it from having a large influence in the posterior samples in the beginning, resulting in more stable optimization.

**Practical optimization with the KL term.**  Experimentally, we find that the ELBO as formulated can be quite often dominated by the KL term during optimization. This is mostly exacerbated when the number of model parameters grows. For this reason, in all experiments, we follow common procedure and instead optimize

$$\mathcal{L}(\hat{\boldsymbol{\theta}}, \sigma_{\mathrm{ker}}, \sigma_{\mathrm{im}}) = \mathbb{E}_{\boldsymbol{\theta} \sim q}[\log p(\mathbf{y}|\boldsymbol{\theta}, \mathbf{x})] - \beta \mathrm{KL}(q(\boldsymbol{\theta})\|p(\boldsymbol{\theta})), \tag{21}$$

where $0 < \beta < 1$ (fixed) controls the relative strength of the KL term.

**Reorthogonalized conjugate gradients (CG).**  As discussed in Section 3.2, we use a variant of conjugate gradients that improves numerical stability by reorthogonalizing the basis vectors at every iteration. Aiming for a good trade off between numerical robustness and performance, we use 10 iterations in all CG solves.

**Posterior sampling.**  For computing the expectation term of the ELBO during training (either while optimizing just sigmas or the full model as well), we use a single sample from our posterior, that is, $S = 1$. Empirically, we observe little gain over using multiple samples and this can be compensated by training for longer, with the benefit of accelerating alternating projections and reducing memory usage significantly. When evaluating on validation and test sets, we draw 20 posterior samples.

## C.1    Architectures and data preprocessing

Unless otherwise noted, all data is standardized, that is, every data point is first subtracted by the mean of training data and the result is divided by the standard deviation of training data, both computed across dimensions. Naturally, this is performed also for the validation and test sets. On MNIST and Fashion MNIST, we train a LeNet; on SVHN and CIFAR-10, we train a small ResNet.

**LeNet architecture.**  Composed of two $5 \times 5$ convolution layers with 6 and 16 channels, respectively, each followed by $2 \times 2$ max-pooling with stride 2, then three fully-connected layers with 128 and 80 units, with the last one being the number of classes as units; all layers have hyperbolic tangent activation functions. The total number of parameters is 44,426.

**ResNet architecture (small variant).**  Composed of $(3, 3, 3)$ residual blocks of channel sizes 16, 32, and 64, respectively; all layers have ReLU activation functions. The total number of parameters is 272,378.

## C.2    Ablation experiments (Section 3)

This section details the experiments illustrated by Figures 2 to 4 in the main paper.

**Stochastic alternating projections.**  For each $\gamma = 0.0, 0.5,$ and $1.0$, we train the LeNet model specified above on the Fashion MNIST dataset. Table 4 details the other hyperparameters used.

**Post-hoc variational inference of $\sigma_{\mathrm{ker}}$ and $\sigma_{\mathrm{im}}$.**  Since we assess the need for a full model ELBO optimization versus only post-hoc tuning of the variational parameters, we need to start from a reasonably-performing model. In both settings, we thus begin with training the model using maximum

| Table 4: Experiments evaluating $\gamma$ (Section 3.3). | |
| --- | --- |
| Hyperparam. | Value |
| $\beta$ | $10^{-4}$ |
| Batch size | 32 |
| Init. $\log \alpha$ | 4 |
| Train. epochs | 50 |
| Learn. rate | $10^{-3}$ |

| Table 5: Post-hoc $\sigma_{\mathrm{ker}}, \sigma_{\mathrm{im}}$ experiments (Section 3.4). | |
| --- | --- |
| Hyperparam. | Value |
| $\beta$ | $10^{-4}$ |
| $\gamma$ | 0.5 |
| Batch size | 32 |
| Init. $\log \alpha$ | 4 |
| Warmup epochs | 20 |
| Learn. rate | $10^{-4}$ |

| Table 6: Warmup experiments (Section 3.4). | |
| --- | --- |
| Hyperparam. | Value |
| $\beta$ | $10^{-4}$ |
| $\gamma$ | 0.5 |
| Batch size | 32 |
| Init. $\log \alpha$ | 4 |
| Learn. rate (MLE) | $10^{-4}$ |
| # epochs (ELBO) | 50 |
| Learn. rate (ELBO) | $10^{-4}$ |

likelihood for 20 epochs, with the same batch size as later, but a learning rate of $10^{-5}$ instead to prevent an early overfitting. Following that, the post-hoc setting optimizes the variational parameters ($\sigma_{\mathrm{ker}}$ and $\sigma_{\mathrm{im}}$) for 20 epochs. In the other setting, in order to draw a better parallel and more clearly highlight the benefits of full model optimization using our ELBO, we tune only the sigmas for 5 epochs. For those epochs, the performance is thus comparable. We then continue with 15 more epochs where the model is also trained, for a total of also 20 epochs. Table 5 shows the values of all other hyperparameters used.

**Warming up as a shortcut.** The only change across different runs is the number of epochs the model was trained with maximum likelihood at the start (warmup): 0, 20, or 50. Table 6 contains the values of the other hyperparameters.

## C.3 Main experiments

The experiments in Section 4 require further details for full reproducibility, given below.

## C.4 Toy (sinusoid) regression

In these experiments, we train a neural network with three fully-connected layers, two with 10 units and one with a single output unit. The training data consists of 10 samples from the sinusoid $y = 5\sin(10x) + z$, where $x \in [0.35, 0.65]$, $z \sim \mathcal{N}(0, s)$, and $s \in [10^{-3}, 1]$, from which 20 consecutive points in specified range of $x$ are drawn, but only the first and last five are used, introducing the visible gap. The VIKING experiment uses the model Jacobian (not loss Jacobian as every other experiment) and a linearized predictive[5] to highlight the properties of guaranteed zero variance in the predictions when projecting onto the kernel of the GGN of the model on training data. The model is trained for 2,000 epochs with a learning rate of $10^{-2}$ and adaptive gradient clipping of rate $10^{-1}$, using 100 posterior samples at each step. The $\log \alpha$ prior ($\log \sigma_{\mathrm{ker}} = -1/2 \log \alpha$, kept tied as described earlier) is initialized with zero. For IVON, we train the same model for 5,000 epochs with the same learning rate, using 5 posterior samples at each training step. The ESS hyperparameter is set to 10 (number of data points) and HESS-INIT is set to 15. The plots in Figure 5 depict the predictions of 100 posterior samples.

## C.5 Image classification

**Grid search.** On MNIST, Fashion MNIST, SVHN, and CIFAR-10, both VIKING and IVON were subjected to a grid search. On VIKING, the grid search is over batch size (16 or 32), number of warmup epochs (Section 3.4; 20 or 50), learning rate ($10^{-3}$ or $10^{-4}$), $\beta$ ($10^{-4}$, $10^{-5}$, or $10^{-6}$), and $\gamma$ (0.2, 0.5, or 0.8). For IVON, the grid search is over batch size (16, 32, or 128), number of epochs (50 or 100), and learning rate ($10^{-3}$ or $10^{-4}$). Table 7 breaks down the best hyperparameters per method and dataset.

---

[5]With $\boldsymbol{\theta} \sim q(\boldsymbol{\theta})$ denoting a posterior sample, instead of making predictions as $\mathbf{y} = f_{\boldsymbol{\theta}}(\mathbf{x})$, we linearize the model around $\hat{\boldsymbol{\theta}}$ as $\mathbf{y} = f_{\hat{\boldsymbol{\theta}}}(\mathbf{x}) + J_{\hat{\boldsymbol{\theta}}}(\mathbf{x})(\boldsymbol{\theta} - \hat{\boldsymbol{\theta}})$.

Table 7: Hyperparameters broken down per dataset. "MC samples" indicates the number of posterior samples used for training.

| | | MNIST | F-MNIST | SVHN | CIFAR-10 |
|---|---|---|---|---|---|
| **VIKING** | Batch size | 32 | 16 | 32 | 32 |
| | Warmup epochs | 50 | 20 | 50 | 20 |
| | Learning rate | $10^{-4}$ | $10^{-4}$ | $10^{-4}$ | $10^{-4}$ |
| | $\beta$ | $10^{-5}$ | $10^{-5}$ | $10^{-6}$ | $10^{-6}$ |
| | $\gamma$ | 0.8 | 0.2 | 0.8 | 0.5 |
| | MC samples | | 1 | | |
| | Warmup learning rate | | $10^{-3}$ | | |
| | Initial $\log \alpha$ | | 4.0 | | |
| | Initial $\log \sigma_{\mathrm{im}}$ | | $-2.0$ | | |
| | $\sigma_{\mathrm{ker}}, \sigma_{\mathrm{im}}$ tuning | | 5 epochs | | |
| | ELBO optimization | | 50 epochs | | |
| **IVON** | Batch size | 16 | 16 | 16 | 16 |
| | Epochs | 100 | 100 | 100 | 100 |
| | Learning rate | $10^{-3}$ | $10^{-3}$ | $10^{-3}$ | $10^{-3}$ |
| | MC samples | | 5 | | |
| | ESS | | Number of data points | | |
| | HessInit | | 1.0 | | |

Table 8: Hyperparameters for the ResNet-34 experiment on Imagenette. "MC samples" indicates the number of posterior samples used for training. In VIKING, epochs and learning rates are split between warmup and ELBO, respectively.

| | VIKING | IVON |
|---|---|---|
| Batch size | 128 | 128 |
| Epochs (warmup, ELBO) | 100, 50 | 150 |
| Learning rate | $5 \cdot 10^{-2}, 10^{-3}$ | $10^{-1}$ |
| $\beta$ | $10^{-7}$ | N/A |
| $\gamma$ | 0.2 | N/A |
| MC samples | 1 | 5 |

**Large models.** In this experiment, we investigate the applicability of VIKING when training a ResNet-34, whose number of parameters is 21,797,672 and compare it against IVON. Due to a limited compute budget with larger models, we do not perform an extensive grid search. Table 8 summarizes all hyperparameters, organized per method.

## C.6 Generative modelling

**Model architecture.** *Encoder:* with five $4 \times 4$ convolutional layers, the first two with 128 channels, followed by the next three with 256 channels, with the first and last having a stride of one and the others a stride of two; two fully-connected layers, one with 256 units and the second with 64 units (latent dimensions) for the VAE approximate posterior mean and 64 for the VAE approximate posterior variance. *Decoder:* two fully-connected layers, one with 256 units and one with 16384 units; five $4 \times 4$ resize convolutional layers (Odena et al., 2016), with the same stride configuration as the encoder, but with 256, 128, 64, and 32 channels, respectively. All layers in the encoder and decoder have an ELU activation function. Total number of parameters: $6.5$ million.

**Maximum likelihood training.** We train for 500 epochs with a maximum learning rate of $10^{-3}$ and batch size 256 on raw data without augmentation. Furthermore, we used the Adamax optimizer and scheduled a 1000-step (updates) linear warmup to the maximum learning rate and then exponentially decayed it.

**VIKING finetuning.**   From the maximum likelihood checkpoint, we train for 10 VIKING epochs using 8 posterior samples, $\gamma = 0.5$, $\beta = 10^{-8}$, batch size 8, learning rate $10^{-4}$ and adaptive gradient clipping at 0.1. The value of $\log \alpha$ is 7.0 and $\log \sigma_{\mathrm{im}}$ is $-5.0$.

**IVON finetuning.**   From the maximum likelihood checkpoint, we train for 20 IVON epochs using 5 Monte Carlo samples, using a batch size of 256 and a learning rate of $10^{-5}$.

