# OpenReview forum: "VIKING: Deep variational inference with stochastic projections"
_NeurIPS.cc/2025/Conference — NeurIPS 2025 poster_

### Official Review · Reviewer_bAN4 · 2025-06-26

**Clarity:** 3
**Significance:** 4
**Originality:** 4
**Rating:** 6
**Confidence:** 5

**Summary:**

This paper proposes a new variational posterior for Bayesian neural networks that has two variance parameters. One parameter controls the scale of the Gaussian distribution in the non-data-affecting directions in parameter space (null-space of the empirical Fisher) and the other controls the scale of a Gaussian distribution in the orthogonal complement subspace.

In order to perform scalable projections into the null space of the empirical Fisher, the authors take the method of alternating projections proposed in Miani et al 2024 and apply it within a conjugate gradient update scheme. To accommodate this in a minibatch setting where the variational posterior and therefore the Fisher changes at every iteration, the authors add an additional approximation where the noise variable has a memory and is updated with a kind of momentum.

The authors perform experiments to validate their method accuracy, uncertainty estimation and scalability. They show an ability to capture in-between uncertainty, and better uncertainty estimation than many competitive BNN inference techniques in image classification with CNNs.

**Questions:**

-	How many CG iterations were used in each optimization step?
-	How does the training wall-clock time compare with IVON and MAP?
-	Suggestion: visualize your approximate posterior in a toy case. It’s quite unusual and it would be good to get more intuition for it.
-	Suggestion: please add pseudocode. There are layers of approximation here: VI, stochastic optimization, CG, alternating projections, stochastic alternating projections. It is quite hard to keep track of the exact steps that need to occur to run the algorithm.
-	Suggestion: could you extend the method to have more than 2 variance parameters? Right now there’s only two kinds of subspace, the null space and its complement. Perhaps there could be ways to partition the space more finely and introduce independent variational parameters for more subspaces.
-	Suggestion: please do not call the method KIVI. There is a name clash with Kernel Implicit Variational Inference: https://arxiv.org/abs/1705.10119

**Ethical Concerns:**

["NO or VERY MINOR ethics concerns only"]

**Final Justification:**

After reading the rebuttals and the other reviews, I maintain my strongly positive view of this paper and greatly hope it is accepted.

**Limitations:**

-	The authors do not explain clearly how much slower their method is.
-	Step-by-step explanation of the method could be clearer.

**Quality:**

4

**Strengths And Weaknesses:**

Quality:

The paper is very well-written and a joy to read. The authors show a great understanding of the context of the field. The experiments are thorough, answering most questions that the reader is interested in, and comparing with good baselines. The authors are clear and informative regarding the limitations of their work, which is refreshing, and avoid over-claiming.

Clarity:

The paper is of above-average clarity, and the major ideas are communicated well in spite of their complexity.

Significance:

Having worked in the BNN field for 7+ years, I would have to say this is one of the top 5 most exciting papers I’ve seen in the field. It is quite rare to have a paper propose not one, but several interesting ideas and string them together this well. Performing good BNN inference is extraordinarily difficult, and the authors are right to note that many in the field believe that performant BNN inference is a lost cause. This work which targets in a perceptive way the exact directions in weight space where approximate posteriors can spread without affecting the loss is significant for breathing new ideas into an old field. Although their method is not “SotA on everything” I think that is almost besides the point in this case. The ideas are fresh and have been demonstrated to be worth pursuing further.

Originality:

The authors note that their work is a continuation of Miani et al 2024 and bears many similarities, although I believe the extension to ELBO optimization during training is significant and non-trivial, and another step in this exciting research direction. I hope to see this avenue developed further in future work.
Note: other methods have proposed subspace inference before, but not in a targeted way, and with a very different inference methodology: https://arxiv.org/abs/1907.07504

---

> ### Author Rebuttal · Authors · 2025-07-30
>
> Thank you for the positive assessment! We're glad you like the paper. We will reply to your questions below.
>
> > How many CG iterations were used in each optimization step?
>
> We used ten CG iterations per optimization step.
>
> > How does the training wall-clock time compare with IVON and MAP?
>
> We summarize the running times (total execution time from model initialization until the end of the last training epoch, in minutes) below, comparing IVON and KIVI:
>
> | Dataset/Architecture  | KIVI           | IVON          |
> |-----------------------|----------------|---------------|
> | MNIST/LeNet           | 27.27 ± 3.96   | 9.97 ± 0.35   |
> | FMNIST/LeNet          | 46.30 ± 7.08   | 9.71 ± 0.06   |
> | SVHN/ResNet_small     | 304.16 ± 0.96  | 108.87 ± 1.90 |
> | CIFAR-10/ResNet_small | 177.08 ± 11.68 | 70.17 ± 0.88  |
>
> As shown by the table, KIVI takes from 2 to 5 times the training time of IVON, where aspects such as alternating projections (and whether the stochastic variant is used) and the number of MLE steps used for warmup can affect the total execution time.
>
> > Suggestion: visualize your approximate posterior in a toy case. It’s quite unusual and it would be good to get more intuition for it.
>
> We have visualizations of the predictive posterior in Figures 1 and 5, which give some indication of the weight posterior. A plot of the weight posterior itself may be a bit tame, as it is merely a sum of two Gaussians projected onto two disjoint subspaces.
>
> > Suggestion: please add pseudocode
>
> We will summarize all the steps in pseudocode in a revised version of the manuscript, together with computational complexity analysis (see the reply to Reviewer rdDS for that one). Thank you for the suggestion.
>
>
> > Suggestion: could you extend the method to have more than 2 variance parameters? Right now there’s only two kinds of subspace, the null space and its complement. Perhaps there could be ways to partition the space more finely and introduce independent variational parameters for more subspaces.
>
> Your suggestion makes a lot of sense, thank you! The variance in the kernel subspace is only informed by the prior (which is usually an isotropic Gaussian), and it ensures that prior variance does not cause the model to underfit. Here a single scalar makes sense, though one could consider a more elaborate prior (e.g., one scalar per layer). But in the image subspace, each basis direction corresponds to different "speeds" along which the loss changes. Here, it would make a lot of sense to consider a more elaborate model, e.g., a low-rank covariance matrix or similar.
>
> > Suggestion: please do not call the method KIVI.
>
> We've also noticed the name clash with the other KIVI since the paper submission and to avoid confusion, we'll rename our method to VIKING ("Variational inference with kernel- and image-subspaces of numerical Gauss--Newton matrices").
>
>
> Again, thanks for your review. We highly appreciate your positive comments and assessment of our work!

---

> ### Comment · Reviewer_bAN4 · 2025-08-05
>
> >We have visualizations of the predictive posterior in Figures 1 and 5, which give some indication of the weight posterior. A plot of the weight posterior itself may be a bit tame, as it is merely a sum of two Gaussians projected onto two disjoint subspaces.
>
> Maybe I'm being silly, but I know in 2D this is trivial (basically and ellipse). In 3D however it's a special kind of ellipse where one axis is "free" and the other two are tied to have identical width. In higher dimensions I don't really have a great intuition for what this looks like at all, i.e. a geometric intuition for the nature of this constraint. Not that it matter too much.
>
> I maintain my score as strong accept. Thank you for answering my questions!

---

> > ### Author Response · Authors · 2025-08-06
> >
> > Your intuitive picture of ellipsoids with equal-length axes is exact. Locally, the fixed-length axes would point in directions where the posterior is constant, which is the key design criterion of our method. We've been thinking more about the suggestion, and it would be quite informative to plot this Gaussian on top of the true posterior (which can be evaluated up to an unknown constant). If we can successfully make such a 3D plot, it would be didactically valuable to include such.
> >
> > Either way, we are grateful for the kind words and strong support.

---

### Official Review · Reviewer_9p1F · 2025-06-30

**Clarity:** 3
**Significance:** 4
**Originality:** 3
**Rating:** 5
**Confidence:** 3

**Summary:**

The paper proposes KIVI, a variational inference method that accounts for overparameterization in deep networks by decomposing the parameter space via the Fisher-Rao metric into kernel and image subspaces. A fully correlated Gaussian posterior with separate variances in each subspace captures structured uncertainty. Using matrix-free stochastic projections and conjugate gradients, KIVI scales to large models. Empirical results show strong performance and improved calibration over baselines like IVON and SWAG.

**Questions:**

- Can $\sigma^2_{\text{ker}}$ and $\sigma^2_{\text{im}}$ be interpreted as epistemic and aleatoric uncertainty in practice?
- In Table 1, KIVI achieves higher accuracy on Imagenette compared to IVON, but with worse calibration metrics. Can you provide an explanation for this trade-off? Does this suggest limitations in the variational family's expressivity?
- IVON has been demonstrated on LLaMA-2 with 7B parameters. Given KIVI's reliance on loss-Jacobian projections and conjugate gradient solves, is it currently feasible to apply your method at that scale?

**Ethical Concerns:**

["NO or VERY MINOR ethics concerns only"]

**Final Justification:**

**Summary of how my questions were addressed**

1. **Runtime & convergence trade-offs vs. IVON**
   Authors provided a runtime table showing KIVI is **2–5× slower** than IVON, explained the main cost factors (loss-Jacobian projections, CG iterations, warmup), and clarified the practical trade-offs.

2. **Interpretation of $\sigma^2_{\text{ker}}$ and $\sigma^2_{\text{im}}$**
   Kernel subspace contains only *epistemic* uncertainty, image subspace contains a *mix* of epistemic and aleatoric uncertainty. Useful as an intuition, but not a strict decomposition.

3. **Feasibility at LLaMA-2 scale**
   Not yet tested at 7B parameters. In theory, scales to that size, but would be 2–5× more expensive than IVON due to projections and CG solves.

**Conclusion:**
All questions were answered with concrete explanations or limitations. I was satisfied with the responses and kept my “Accept” score.

**Limitations:**

yes

**Paper Formatting Concerns:**

no issues

**Quality:**

3

**Strengths And Weaknesses:**

### **Strengths**

- The method is grounded in a principled decomposition of the parameter space using the Fisher-Rao metric
- Uses a minimal parameterization while still achieving competitive performance
- Scalable to large models through matrix-free methods
- Empirically outperforms or matches state-of-the-art methods

### **Weaknesses**

- Lacks theoretical analysis or guarantees on convergence and approximation quality
- The paper does not provide any runtime or convergence comparisons between KIVI and IVON. While KIVI introduces additional computational overhead due to conjugate gradient-based projections, there are no wall-clock benchmarks or convergence plots to quantify the practical trade-offs.

---

> ### Author Rebuttal · Authors · 2025-07-30
>
> Thank you for the valuable review! In the following, we answer the questions individually.
>
> > The paper does not provide any runtime...
>
> We summarize the running times (total execution time from model initialization until the end of the last training epoch, in minutes) below, comparing IVON and KIVI:
>
> | Dataset/Architecture  | KIVI           | IVON          |
> |-----------------------|----------------|---------------|
> | MNIST/LeNet           | 27.27 ± 3.96   | 9.97 ± 0.35   |
> | FMNIST/LeNet          | 46.30 ± 7.08   | 9.71 ± 0.06   |
> | SVHN/ResNet_small     | 304.16 ± 0.96  | 108.87 ± 1.90 |
> | CIFAR-10/ResNet_small | 177.08 ± 11.68 | 70.17 ± 0.88  |
>
> As shown by the table, KIVI takes from 2 to 5 times the training time of IVON, where aspects such as alternating projections (and whether the stochastic variant is used) and the number of MLE steps used for warmup can affect the total execution time.
>
> > Can $\sigma^2_{\text{ker}}$ and $\sigma^2_{\text{im}}$ be interpreted...
>
> While we initially were optimistic that this would be the case, we eventually avoided this interpretation. The kernel contains only epistemic uncertainty, while the image can contain both epistemic and aleatoric uncertainty. In practice, we observe that the image space does contain a mix of uncertainties, and, depending on model complexity, this changes in degree. So, while the interpretation is a useful starting intuition, it should be followed with some care.
>
> > In Table 1, KIVI achieves higher accuracy on Imagenette ...
>
> That's a good question. The calibration metrics are relatively similar (given the standard deviations), so concluding limitations on the expressivity may be a slight overinterpretation of these results. For example, on CIFAR-10 and SVHN, KIVI's calibration is better than IVON's.
>
> > IVON has been demonstrated on LLaMA-2 with 7B parameters...
>
> Good point. We haven't explored that scale for KIVI yet. You're right in that the loss-Jacobian projections and conjugate gradient solves make KIVI more expensive than IVON. Even though KIVI scales well, it costs about 2-5 times the training time of IVON; see our complexity analysis in the reply to Reviewer rdDS, which we will also add to the paper.
>
> Again, thanks for your valuable review! We hope that all concerns have been resolved adequately.

---

> > ### Comment · Reviewer_9p1F · 2025-08-03
> >
> > Thank you for your clarifications. I will maintain my score.

---

### Official Review · Reviewer_4Nga · 2025-07-01

**Clarity:** 2
**Significance:** 1
**Originality:** 1
**Rating:** 5
**Confidence:** 3

**Summary:**

This paper introduces a novel approach for Gaussian variational inference in the context of over-parameterized Bayesian neural networks. Motivated by the observation that neural network outputs remain invariant within the kernel of the Fisher information matrix, and building on recent work by Miani et al. (2024b), the authors propose a new form of approximate Gaussian posterior distribution. The covariance matrix of this distribution is constructed as a combination of two components: $UU^\top$ and $I-UU^\top$.
Samples from the range of the former (i.e., the kernel of the Fisher matrix) do not affect the network’s prediction on the training data, whereas samples from the latter do. To reduce the computational cost of estimating the Fisher information matrix, the authors adopt a stochastic projection technique from Miani et al. (2024b). The proposed method is evaluated on out-of-distribution (OOD) detection and variational autoencoder tasks, demonstrating its effectiveness.

**Questions:**

Miani et al. (2024b) propose restricting the approximate posterior to the kernel of the Fisher-Rao matrix, ensuring that samples do not alter the model's predictions on the training data. This makes intuitive sense: it maintains predictive accuracy on the training data while allowing the model to express uncertainty on out-of-distribution (OOD) inputs.

Your method, in contrast, allows posterior mass in both the kernel $R\left(U U^{\top}\right)$ and its complement, even though the latter influences the model's predictions on the training set. Could you clarify the motivation for this choice? How does introducing variability in directions that change the training predictions contribute to the overall performance or uncertainty calibration? I request an explanation that is more than simply showing me the experimental results.

**Ethical Concerns:**

["NO or VERY MINOR ethics concerns only"]

**Final Justification:**

I appreciate the author's response, which has addressed most of my concerns. I now increase my score to accept.

**Limitations:**

Yes.

**Quality:**

1

**Strengths And Weaknesses:**

1. The paper lacks sufficient discussion of prior work on Gaussian variational inference for Bayesian neural networks. While this is a broad area, some key references should be acknowledged, at least in the appendix. Here is an inexaustive list: the linearized Lapalce approximation by https://arxiv.org/abs/2008.08400, https://arxiv.org/abs/1906.01930; the functional space variational inference by https://arxiv.org/abs/2312.17199. I am not asking the authors to empriicaly compare with these papers, but at least these closely related work should be properly discussed.

2. The author is not distinguishing between the expected value of the Fisher information matrix and its empirical estimate over the training samples. I think the author is using the latter in the experiment. Since the experiments likely use the empirical version, it is important to clarify whether the desirable structural properties (e.g., predictive invariance within the kernel) still hold under this approximation.

3. To improve clarity, I recommend adding a subscript to the matrix $U$, such as $U_{\hat{\theta}}$, to emphasize that it is computed at a specific parameter point $\hat{\theta}$. This is particularly important in Section 3.3, where the dependence of the Fisher kernel on the parameter estimate is critical to the algorithm.

4. While I generally avoid making novelty the primary critique, in this case it is a significant concern. Much of the methodology, including the design of the covariance matrix and the stochastic projection trick, appears to be directly adapted from Miani et al. (2024b). The key difference-that this work uses a linear combination of $U U^{\top}$ and $ \mathrm{Id}-U U^{\top}$ rather than restricting the posterior entirely to the kernel-is incremental and may not constitute a substantial enough contribution to justify publication. Given the heavy reliance on prior work, I do not believe the paper introduces sufficient novel insights, and I therefore recommend rejection.

---

> ### Author Rebuttal · Authors · 2025-07-30
>
> Thank you for the valuable feedback! In the following, we'll briefly reply to what the review lists under weaknesses, before answering the specific questions.
>
> > The paper lacks sufficient discussion of prior work on Gaussian variational inference for Bayesian neural networks.
>
> Thank you for pointing out relevant work we overlooked in our discussion. A reviewed version of the manuscript will appropriately mention those papers.
>
>
> > The author is not distinguishing between the expected value of the Fisher information matrix and its empirical estimate over the training samples.
>
> Thank you for raising this point. In this paper, we target the kernel and image spaces of the Generalized Gauss-Newton matrix. The Fisher information matrix i.e.
> $\sum_n  \mathbb{E}\_{y \sim p(y|x\_n)}  [\nabla_\theta \log p(y|x\_n) \nabla_\theta \log p(y|x_n)^T]$
> corresponds exactly to the generalised Gauss-Newton matrix in our setting; see Proposition 1 by Kunstner ("Limitations of the Empirical Fisher Approximation for Natural Gradient Descent", 2019). Note that this is distinct from the empirical Fisher i.e. $\sum_n  [\nabla_\theta \log p(y_n|x_n) \nabla_\theta \log p(y_n|x_n)^T]$. So, we rely on empirical estimates over the training data, as is standard in deep learning, because we don't know the distribution over inputs, but we don't rely on empirical estimates over the training data and the labels as used in the empirical Fisher. We retain all the desirable structural properties of the loss-projected posterior from Miani 2024b since we use the same matrix.
>
> > To improve clarity, I recommend adding a subscript to the matrix...
>
> We agree, thank you for the feedback. The notation around $U$ and its dependence on the current parameter point estimate is now explicit in the manuscript.
>
> > While I generally avoid making novelty the primary critique, in this case it is a significant concern. Much of the methodology, including the design of the covariance matrix and the stochastic projection trick, appears to be directly adapted from Miani et al. (2024b).
>
> Thank you for that comment.
> Please note that our work has several crucial differences from Miani et al. (2024b). Miani et al. (2024b) is a post hoc method, which means they sample from the proposed approximate posterior after MAP training. Our work constructs a variational family based on the approximate posterior proposed. While we borrow techniques for efficiently sampling from the variational family, the full ELBO training is a significant innovation that clearly distinguishes our contribution from Miani et al. (2024b).
>
> **The last point relates to the answer to your question. To elaborate on that:**
>
> As stated by the reviewer, the kernel directions in the parameter space are where the predictions are locally constant. In Miani et al. (2024b), this property is exploited to build posteriors with 0 in-distribution uncertainty but high out-of-distribution uncertainty, leading to good OOD detection. However, since we propose a method for variational training, it is important that we converge to a distribution that has a good predictive performance and sensible uncertainty quantification. This is why it is crucial to include the image component in the posterior because that is the subspace where the predictions (and the loss) actually change (locally).
>
> Again, thanks for the thorough review! We hope that all concerns have been addressed, and we're looking forward to discussing.

---

> > ### Comment · Reviewer_4Nga · 2025-08-01
> > **Thanks for the response**
> >
> > I appreciate the author's response, which has addressed most of my concerns. I now increase my score to accept.

---

### Official Review · Reviewer_rdDS · 2025-07-02

**Clarity:** 3
**Significance:** 2
**Originality:** 2
**Rating:** 3
**Confidence:** 3

**Summary:**

This paper proposes KIVI, kernel image variational inference, which is a full rank Gaussian posterior approximation with a covariance that is a full rank basis constructed around the pseudo inverse of the Jacobian. Experiments are performed on toy regression tasks as well as smaller image based deep nets.

**Questions:**

I believe that the first use of conjugate gradients to solve systems of the form $(J’J)^{-1}J’v$ in a matrix free fashion is from Maddox et al, ’21 (https://arxiv.org/pdf/2103.01439), which does yield an interesting question in terms of the relationship between the posterior approximation and the learnable local Laplace approximations (e.g. Immer et al, ’20 [https://arxiv.org/abs/2008.08400], Antoran et al, ’22 [https://arxiv.org/abs/2210.04994] beyond just the ones cited in your work). In general, there’s a pretty deep connection to the first order Taylor approximation / linearization of the neural network and thus things like the local Laplace approximation and the neural tangent kernel.

 I would generically think that your approach boils down to training a predictive mean through something like the linearization in Maddox et al, ’21 rather than simply basing the covariance estimate off of a pre-trained model like they did. I wonder if the authors could connect their work to these types of linearizations a bit more?

Eq 2 – why do the authors not instead optimize $\Sigma = \sigma_K^2 UU’ + \sigma_n I$ as $\sigma_K = (\sigma_{ker}^2 - \sigma_{im}^2)$ in your parameterization and we need to ensure that the covariance is psd so $\sigma_K^2 >= 0$ by construction.

L147 / Eq 13: for further scalability, why not use SGD (or a cheap second order) method to minimize that loss (possibly with the relaxed constraint that \lambda is small enough)? This should only require Jacobian vector products instead of Jacobian matrix solves..

Eq 16: P_t is not really defined previously. What is the “kernel-projection of the t th 168 batch”? Is this the reorthogonalization matrix from the previous solve?

How accurate (in terms of forwards error) are the CG solves after only 10 steps?

Figure 5: why are the data points different for IVON and KIVI?

Tables 1 and 2: I’m generally a bit surprised at the high ECE values as these are much higher than whats reported in Table 3 of Maddox et al, ’19 (the original SWAG paper) but on checking Table 2 of https://arxiv.org/pdf/2410.16901 I see that they are the same. Could the authors comment on the difference (I believe that this is architectural but would like confirmation)?

Writing comments:

-	L160:  “- we found.. “ turn into a new sentence.

**Ethical Concerns:**

["NO or VERY MINOR ethics concerns only"]

**Final Justification:**

I thank the authors for responding to the comments during the rebuttal period. Overall, I still think that the paper is quite borderline and further discussion with prior work (as I pointed out in the original review) and a comparison with deep ensembles/sgmcmc (rather than just vi methdods) is needed for acceptance.

**Quality:**

3

**Strengths And Weaknesses:**

Strengths:

The method is well described including all of the computational tricks that are needed to make it work.

Weaknesses:

No discussion of the computational expense, which I believe should be quite expensive due (at a minimum 3x backwards * the number of CG steps per backwards passes). Scaling in my mind is pretty important here as it prevents doing some interesting experiments here either using the probabilistic predictions somehow or in doing optimization from large pretrained models.

I believe that Appendix A.1 yields a slightly sub-optimal gradient estimate, which should probably be discussed in the main text.

The missing comparison to me is against a low rank (say rank 10) plus diagional (SLANG I believe, https://arxiv.org/abs/1811.04504) variational Gaussian posterior that just directly optimizes U instead of trying to structure it heavily. Additionally, I would probably argue for a MCMC based comparison like SGLD / SGHMC as those tend to be relatively cheap to run as well.

---

> ### Author Rebuttal · Authors · 2025-07-30
>
> Thank you for that thoughtful assessment. In the following, we will first reply to the weaknesses before answering the specific questions.
>
> > No discussion of the computational expense, which I believe should be quite expensive due (at a minimum 3x backwards * the number of CG steps per backwards passes). Scaling in my mind is pretty important here as it prevents doing some interesting experiments here either using the probabilistic predictions somehow or in doing optimization from large pretrained models.
>
>  Thanks for bringing this up. We agree that providing more explanation of the complexity would improve the presentation. We've now added the following paragraph to highlight the linear dependency on every hyperparameter in our method and to clarify the total computational complexity.
>
> > ### Proposed changes:
> > **Computational complexity.** Assume a Jacobian-vector-product and a vector-Jacobian product through the model function at a single data-point each cost $\mathcal{O}(D)$. Then, for batch-size $B$, the least-squares solver needs $BN_{\text{lstsq}}$ of these matrix-vector and vector-matrix products to implement the projections. For each of the $S$ Monte Carlo samples (Equations 8-12) samples, a standard-Gaussian sample ${\epsilon}^{(s)}$ is turned into a kernel-projection sample ${\epsilon}\_{\text{ker}}^{(s)}$ in complexity $\mathcal{O}(B N\_{\text{lstsq}} D)$. Afterwards, all $S$ kernel-projection samples are linearly combined to estimate the reconstruction-term $\mathbb{E}\_{{\theta} \sim q}\left[ \log p(y | {\theta}, x) \right]$ and the KL-term $\mathrm{KL}( q({\theta}) \| p({\theta}) )$. Therefore, a single ELBO-evaluation costs $\mathcal{O}(SBN\_{\text{lstsq}} D)$. Our experiments solve the least-squares problem using $N_{\text{lstsq}}=10$ Jacobian-vector and vector-Jacobian products, varying batch-sizes, and $S=1$ Monte Carlo samples. See Appendix B for details.
>
> We summarize the running times (total execution time from model initialization until the end of the last training epoch, in minutes) below, comparing IVON and KIVI:
>
> | Dataset/Architecture  | KIVI           | IVON          |
> |-----------------------|----------------|---------------|
> | MNIST/LeNet           | 27.27 ± 3.96   | 9.97 ± 0.35   |
> | FMNIST/LeNet          | 46.30 ± 7.08   | 9.71 ± 0.06   |
> | SVHN/ResNet_small     | 304.16 ± 0.96  | 108.87 ± 1.90 |
> | CIFAR-10/ResNet_small | 177.08 ± 11.68 | 70.17 ± 0.88  |
>
> As the table shows, KIVI takes from 2-5 times the training time of IVON, where aspects such as alternating projections (and whether the stochastic variant is used) and the number of MLE steps used for warmup can affect the total execution time.
>
> > I believe that Appendix A.1 yields a slightly sub-optimal gradient estimate, which should probably be discussed in the main text.
>
> You are correct that not differentiating through the rank estimation leads to an inexact gradient. We opted to stop the gradient because the rank itself is not differentiable (only its estimate is), and we expected instabilities. We will emphasize this inexactness. We have investigated whether this choice matters, and found that it made no difference whether we differentiate through the rank estimation or not.
>
> > The missing comparison to me is against ... [SLANG/MCMC]
>
> Thank you for the suggestions. While our focus has been on methods that are generally considered state-of-the-art for BNNs, we agree that these are interesting baselines, with the following reservations:
> * The general experience is that 'regular' variational methods for BNNs are quite unstable, leading to either underfitting or overfitting depending on initialization and choice of hyperparameters. For mean field approaches, this experience has been the driving motivation for IVON and similar methods. In our experience, SLANG suffers from similar issues, and we worry that reporting such results would not be informative, as highly different results could easily be obtained by changing initialization/hyperparameters. We are open to including such results, but they would come with notable caveats.
> * While SGMCMC methods are cheap to run, they can be quite difficult to tune and may not converge to reasonable samples [1, 2]. For this reason, SGMCMC methods are not commonly used for BNNs. As above, we are open to including results obtained with SGMCMC, but similar caveats would apply.
>
> [1] Bardenet, Rémi, Arnaud Doucet, and Chris Holmes. "On Markov chain Monte Carlo methods for tall data." Journal of Machine Learning Research 18.47 (2017): 1-43.
>
> [2] Brosse, Nicolas, Alain Durmus, and Eric Moulines. "The promises and pitfalls of stochastic gradient Langevin dynamics." Advances in Neural Information Processing Systems 31 (2018).
>
>
> **Next, we reply to the direct questions:**
>
> > I believe that the first use of conjugate gradients...
>
> Yes, these are indeed related. The submission cites linearized Laplace approximations in Section 2. We'll add the papers you list in this section as we agree they should be mentioned.
>
> > I would generically think that your approach boils down to training a predictive mean ...
>
> That is correct, we train (without MLE warm-up) from scratch using the ELBO. Aside from the differences in the covariance structure from our choice of variational family, concretely, we optimize the reconstruction (expectation) part of the ELBO using linearization around the current posterior mean, with the posterior samples parametrizing the linearized model. Thus, the posterior mean (when optimized with the ELBO) is allowed to be influenced by the covariance and not simply optimized in isolation with UQ done post-hoc, since the image component would also allow predictions to deviate from the mean predictions under the training data.
>
> > Eq 2 – why do the authors not instead optimize ...
>
> Thank you for that suggestion. We keep $\sigma_\text{ker}$ and $\sigma_\text{im}$ separate to give different weights to image- and kernel-spaces, respectively. In the current parametrisation, both $\sigma_\text{ker}^2 U U^\top$ and $\sigma_\text{im}^2(I - U U^\top)$ are PSD (as projection matrices), so their sum is PSD. Does that help?
>
> > L147 / Eq 13: for further scalability, why not use SGD (or a cheap second order) method to minimize that loss (possibly with the relaxed constraint that \lambda is small enough)? This should only require Jacobian vector products instead of Jacobian matrix solves..
>
> Thanks for that suggestion! We'd like to emphasise that Equation 13 has a closed-form solution, hence we use conjugate gradients, not SGD. One could attempt to minimise the loss with something like SGD, but it's unclear whether it improves scalability (solving the system with 10 CG steps vs minimising with 10 SGD steps has similar complexity), and we need to ensure that the solves are of high quality. When we lose orthogonality of samples, training becomes very unstable. Also, differentiating through full SGD runs tends to be rather slow.
>
>
> > Eq 16: P\textsubscript{t} is not really defined previously...
>
> We apologize for the confusion caused by the notation. We have updated the text in line 168 to explicitly mention that $P^{(t)} = {U}{U}^{\top(t)}$, where $t$ denotes Eq. 13 restricted for the $t$-batch (w.r.t. $\mathbf{J}_{{\theta}}$). Following the suggestion of another reviewer, we also include $\hat{\theta}$ as a subscript of $U$ to make it explicit that it depends on the current posterior mean.
>
> > How accurate (in terms of forwards error) are the CG solves
>
> We have normalized the forward error with respect to the norm of the r.h.s. (the $b$ in the $Ax = b$ sense), which we refer to as "residuals" and summarized the average residual across different runs for the last epoch:
>
> | Dataset/Architecture  | KIVI            |
> |-----------------------|-----------------|
> | MNIST/LeNet           | 0.0750 ± 0.0295 |
> | FMNIST/LeNet          | 0.0104 ± 0.0021 |
> | SVHN/ResNet_small     | 0.0096 ± 0.0003 |
> | CIFAR-10/ResNet_small | 0.0317 ± 0.0047 |
>
> As evidenced above, the errors are quite acceptable, and the performance observed in the other metrics confirms this as well. Relatedly, Gaussian process software uses coarse conjugate-gradients approximations for covariance matrices, and successfully so (see, e.g., the GPyTorch documentation regarding conjugate gradients).
>
>
> > Figure 5: why are the data points different for IVON and KIVI?
>
> The data points in Figure 5 are the same, but the y-axes are scaled differently. Thanks for pointing this out. We have updated Figure 5 so that both y-axes have the same scale.
>
> > Tables 1 and 2: I'm generally a bit surprised at the high ECE values ...
>
> Yes, that is correct, the ECE numbers are not comparable because of a difference in architecture and hyperparameters. In our experiments, we use the recommended hyperparameters from Maddox et al, ’19.
>
> > Writing comment
>
> Fixed, thanks!
>
> Again, thank you for your review. We hope that all concerns have been resolved; if not, please let us know!

---

> ### Comment · Reviewer_rdDS · 2025-08-02
>
> Thank you for the detailed response, a few more comments and questions:
>
> >KIVI takes from 2-5 times the training time of IVON,
>
> What is the relationship of IVON training time to something like a similar stochastic gradient step? From a quick skim, it seems like IVON should also be much slower.
>
> > methods that are generally considered state-of-the-art for BNNs
>
> I would argue that SGMCMC methods are at least near state of the art for BNNs as they scale well and preserve pretty high accuracy, especially with multiple chains. This is true even if they are not precisely reflective of the "true posterior". See for example the discussions in [Izmailov et al, '21](https://proceedings.mlr.press/v139/izmailov21a/izmailov21a.pdf), [Pituk et al, '25](https://openreview.net/pdf?id=x5RQnF7Vw9) amongst others... Thus, I think it is a pretty clear baseline, especially as it lies on a different position on the accuracy / computational cost frontier than your method.
>
> > $\sigma_k UU^\top + \sigma_\text{im}^2(I - U U^\top) = \sigma_\text{im}^2 I + (\sigma_k - \sigma_\text{im}) UU^\top)$ are PD
>
> I think im not totally sure as towhy the sum of the two matrices is going to stay PD. The parameterization I pointed out stays PD by construction. The concern is primarily if $ (\sigma_k - \sigma_\text{im}) < 0$ then I'm not totally sure what happens.
>
> > forwards error
>
> Is this relative error or absolute error?
>
> > it's unclear whether it improves scalability
>
> Generally, this is a good point; thanks for the clarification. I do think you can do similar tricks for CG and SGD here - an adjoint style approach, so differentiation isn't slower.
>
> > related work of ntk + Jacobian solves
>
> I brought these up because they point out some interesting properties of the variational posterior that you learned as well. Thus, I think it's not that they should be cited, but a bit of discussion is good. For example, the ntk literature suggests that weights don't really move that much for wide enough networks - in your variational posterior does the jacobian change during training from scratch at all?

---

> > ### Author Response · Authors · 2025-08-04
> > **Thanks for your engagement**
> >
> > > What is the relationship of IVON training time to something like a similar stochastic gradient step?
> >
> > The IVON optimization steps are slightly more expensive than Adam, but it does not have much impact on the total optimization time, which is dominated by gradient computations; see Figure 1 in the paper by Shen et al. (2024), which includes wall-clock comparisons of IVON and Adam.
> >
> > > I would argue that SGMCMC methods are at least near state of the art for BNNs as they scale well and preserve pretty high accuracy,
> >
> > We do not disagree that SGMCMC methods can work well, but we generally experience their performance to be overly sensitive to hyperparameters. This makes it difficult to sensibly report their performance in a paper. That being said, we can add such a baseline to the paper. It is, however, unclear if we will be able to provide results within the discussion period.
> >
> > Should the paper be accepted, we are happy to include such a baseline for reference with the noted caveat about hyperparameter-tuning.
> >
> > > I think im not totally sure as to why the sum of the two matrices is going to stay PD
> >
> > The covariance in question takes the form, $\Sigma = \sigma_k^2 U U^{\top} + \sigma_{im}^2 (I - U U^{\top})$. Here, $U U^{\top}$ is a rank $R$ projection matrix, such that it has $R$ eigenvalues that are 1 and $D-R$ eigenvalues that are 0. The matrix $I - U U^{\top}$ is likewise a projection matrix that has $D-R$ eigenvalues that are 1 and $R$ eigenvalues that are 0. The eigenvectors of $U U^{\top}$ and of $(I - U U^{\top})$ are, by definition, orthogonal, such that together they span all $\mathbb{R}^D$. A positive weighted combination of $U U^{\top}$ and $(I - U U^{\top})$, thus, has rank $D$, i.e., full rank. The eigenvalues of $\Sigma$ will then be $\sigma_k^2$ (repeated $R$ times) and $\sigma_{im}^2$ (repated $D-R$ times).
> >
> > > Is this relative error or absolute error?
> >
> > The table displays the absolute error, scaled by the norm of the right-hand side to ensure comparability across runs: $\text{error} = \|Ax-b\|_2 / \|b\|_2$.
> >
> > > in your variational posterior does the jacobian change during training from scratch at all?
> >
> > Yes, when we train from scratch, the Jacobian changes significantly. We observe a larger change in the beginning of training.

---

> > > ### Comment · Reviewer_rdDS · 2025-08-04
> > >
> > > > The eigenvalues of $\Sigma$ will then be $\sigma_k^2$ (repeated $R$ times) and $\sigma_{im}^2$ (repated $D-R$ times).
> > >
> > > thanks, i was able to numerically verify this. Sorry for the confusion. I do think that my parameterization is a bit more natural however.
> > >
> > > > see Figure 1 in the paper by Shen et al. (2024), which includes wall-clock comparisons of IVON and Adam.
> > >
> > > It would be good to include the Adam comparison here as that's the more natural comparison. I'm assuming from that plot the MNIST comparison would be something like 27 / 10 / 7 and with similar scaling on the other problems (so that Adam is about another 30% faster).
> > >
> > > > We observe a larger change in the beginning of training.
> > >
> > > Thanks for the clarification.

---

> > > > ### Author Response · Authors · 2025-08-04
> > > >
> > > > > thanks, i was able to numerically verify this. Sorry for the confusion. I do think that my parameterization is a bit more natural however.
> > > >
> > > > Yes, we acknowledge that there are many equivalent ways of parametrizing this covariance as a function of U and two different scales. However, the objective of our paper is to separate the covariance matrix into kernel and image subspaces with different scales associated to them. This is directly achieved by the parameterization we choose.
> > > >
> > > > > It would be good to include the Adam comparison here as that's the more natural comparison. I'm assuming from that plot the MNIST comparison would be something like 27 / 10 / 7 and with similar scaling on the other problems (so that Adam is about another 30% faster).
> > > >
> > > > We'd be happy to include Adam wall clock comparison alongside IVON in the final version of the paper.
> > > >
> > > > Thank you for following up and engaging in a fruitful discussion. We hope all the issues have been addressed and would appreciate if you'd consider revising your score.

---

> ### Comment · Reviewer_bAN4 · 2025-08-05
>
> I would simply like to add that in my opinion a comparison with IVON and Adam is good enough, and that an SGMCMC comparison, although nice to have, would not be completely necessary to recommend acceptance. I believe that they have taken one method on the "frontier" of VI (IVON) and shown competitive performance with a very different approach while staying in the VI family. That is not enough to claim superiority over all BNN inference methods, but I believe it is enough to claim significance and be worthy of conference publication.

---

> ### Comment · Reviewer_rdDS · 2025-08-06
>
> >  my opinion a comparison with IVON and Adam is good enough, and that an SGMCMC comparison, although nice to have, would not be completely necessary to recommend acceptance.
>
> I don't know how I feel about this. Reading the IVON paper (https://arxiv.org/pdf/2402.17641) i also don't see a SGMCMC comparison (which feels deeply strange to me)... It seems like ensembles are at least more highly performant (e.g. in Table 2), and maybe that's just the better baseline. 5x slower per iteraion is approximately the cost of a standard training ensemble of size 5 for example.

---

> > ### Comment · Reviewer_bAN4 · 2025-08-07
> >
> > Agreed that a five-fold ensemble would be a sensible and straightforward to implement baseline here

---

> > > ### Author Response · Authors · 2025-08-07
> > >
> > > Thank you for sharing this perspective. While we consider our experimental work sufficient to demonstrate our method compared to similar ones, we acknowledge your point that a wider comparison would be interesting. With that in mind, we'd like to offer including both SGMCMC and ensemble baselines should the paper be accepted.

---

### Official Review · Reviewer_boCo · 2025-07-02

**Clarity:** 2
**Significance:** 3
**Originality:** 2
**Rating:** 5
**Confidence:** 3

**Summary:**

This paper is motivated by the insufficiency of existing solutions in approximate inference for overparameterized deep neural networks. Aiming to address the pathologies of such methods in model training, predictive performance and calibration, the authors propose a novel variational family that decomposes into two orthogonal subspaces in the parameter space, spanning the in- and out-of training data supports. After developing a copious iterative algorithm to compute the corresponding posterior approximation, they demonstrate empirically that the novel scheme can often achieve higher predictive power and uncertainty quantification compared to state-of-the-art baselines across toy and image datasets in regression, classification and generative modeling tasks.

**Questions:**

- Projection matrix $U$ seems to be pre-maturely introduced into the notation and downplayed over experiments. I would recommend explaining it further and seeking for further insights in it over the experiments. Can it leverage potential sparsity on a data- and model-specific basis in practice?
- It would be nice to see the effects of the data geometry on the efficacy of the VI method. Is it crucially benefited by clean-cut manifolds, as possibly the ones arising in image data? How would UQ behave in blurred versions of the images?
- Practitioners and experts would benefit from a more thorough discussion of the computational complexity of the method, and potentially a time-comparison against considered baselines across the experiments.
- The new method does not consistently outperform baselines in predictive and calibration performance. How are these findings explained? What makes SVHN a better benchmark to demonstrate the advantages of the methods compared to MNIST and Fashion MNIST?
- A discussion on limitations of extending the method towards language models would help the reader get a clearer understanding of the scope and generalizability of the work.

**Ethical Concerns:**

["NO or VERY MINOR ethics concerns only"]

**Final Justification:**

After reading the authors' response, I am increasing my final score to "Accept." I appreciate the technical contribution of the work on overparameterized Bayesian Neural Networks and believe it meets the conference's standards for evaluation, reproducibility, and ethical considerations. I am agreeable to the authors' proposed edits regarding the discussion of computational complexity, though I believe it would be appropriate to include them in the appendix.

**Limitations:**

Yes

**Paper Formatting Concerns:**

No formatting concerns raised.

**Quality:**

3

**Strengths And Weaknesses:**

**Strengths**

The writing of the paper is mostly clear. It begins with an engaging introduction to the problem, that offers some useful information-geometric intuition underneath the emerging pathologies in uncertainty quantification via approximate inference methods in overparameterized models. This leads to a clear motivation of why the examined problem is significant and paves the way towards the development of the proposed new variational family. Despite appearing as a simple contribution that relies on a fundamental idea of orthogonal decomposition and a minimalistic number of learnable variational parameters, Kernel-Image VI offers clear geometric interpretability. Moreover, it entails an involved computational scheme and can be efficiently combined with warm-up MLE-based training stages offering uncertainty quantification as an add-on on scalable training methods.

**Weaknesses**

The paper has several areas that could be clarified and expanded to better support its contributions$-$consult also the following section on Questions. Notably, the projection matrix $U$ is introduced early in the formulation but is insufficiently motivated and receives limited attention in the experimental analysis. Additionally, the impact of data geometry on the proposed variational inference method remains underexplored. It would be informative to assess how the method performs under varying levels of manifold clarity, such as comparing clean-cut image structures with their blurred counterparts. The computational complexity of the approach is also not addressed in detail; a comprehensive analysis, including empirical runtime comparisons with baseline methods, would be beneficial for practitioners. Furthermore, the proposed method does not consistently surpass existing baselines in terms of predictive accuracy or calibration, and the reasons for these findings are not adequately discussed. Lastly, the manuscript would benefit from a critical examination of the limitations involved in extending the methodology to language models, thereby providing a clearer perspective on its generalizability.

---

> ### Author Rebuttal · Authors · 2025-07-30
>
> We thank the reviewer for the thoughtful comments. Below, we will reply to each point individually.
>
> > Projection matrix \(U\) seems to be pre-maturely introduced into the notation and downplayed over experiments. I would recommend explaining it further and seeking for further insights in it over the experiments. Can it leverage potential sparsity on a data- and model-specific basis in practice?
>
>
> Thanks for raising this point. (Please note that the projection matrix onto the kernel (null) space is $U U^\top$.) What we are interested in, based on existing theoretical developments on model reparametrization, is to project parameter vectors onto the kernel space of the model GGN, as it provides then a clear space of parameters which follow certain properties, e.g., maintain model predictions on data points used to build the stacked Jacobian $\mathbf{J}_{{\theta}}$. Indeed, any solution (or approximation) to Eq. 13 would be particularly interesting, as it would follow those properties, but without instantiating the projection matrix, since it is not tractable.
>
>
> > It would be nice to see the effects of the data geometry on the efficacy of the VI method. Is it crucially benefited by clean-cut manifolds, as possibly the ones arising in image data? How would UQ behave in blurred versions of the images?
>
> We do not see reasons for our method to be dependent on data geometry or a specific task as we are focusing on uncertainty in overparametrized models. Thus, the amount of uncertainty will not influence the ability of our method to detect it. In other words, if the nature of the data leads to a higher uncertainty, we certainly expect this to be reflected in the $\sigma^2_{\text{ker}}$ and $\sigma^2_{\text{im}}$ parameters.
>
> > Practitioners and experts would benefit from a more thorough discussion of the computational complexity of the method, and potentially a time-comparison against considered baselines across the experiments.
>
> We propose adding the following paragraph to highlight the linear dependency on every hyperparameter in our method and clarifying the total computational complexity.
>
> > ### Proposed changes:
> > **Computational complexity.** Assume a Jacobian-vector-product and a vector-Jacobian product through the model function at a single data-point each cost $\mathcal{O}(D)$. Then, for batch-size $B$, the least-squares solver needs $BN_{\text{lstsq}}$ of these matrix-vector and vector-matrix products to implement the projections. For each of the $S$ Monte Carlo samples (Equations 8-12) samples, a standard-Gaussian sample ${\epsilon}^{(s)}$ is turned into a kernel-projection sample ${\epsilon}\_{\text{ker}}^{(s)}$ in complexity $\mathcal{O}(B N\_{\text{lstsq}} D)$. Afterwards, all $S$ kernel-projection samples are linearly combined to estimate the reconstruction-term $\mathbb{E}\_{{\theta} \sim q}\left[ \log p(y | {\theta}, x) \right]$ and the KL-term $\mathrm{KL}( q({\theta}) \| p({\theta}) )$. Therefore, a single ELBO-evaluation costs $\mathcal{O}(SBN\_{\text{lstsq}} D)$. Our experiments solve the least-squares problem using $N_{\text{lstsq}}=10$ Jacobian-vector and vector-Jacobian products, varying batch-sizes, and $S=1$ Monte Carlo samples. See Appendix B for details.
>
> We additionally summarize the running times (total execution time from model initialization until the end of the last training epoch, in minutes) below, comparing IVON and KIVI:
>
> | Dataset/Architecture  | KIVI           | IVON          |
> |-----------------------|----------------|---------------|
> | MNIST/LeNet           | 27.27 ± 3.96   | 9.97 ± 0.35   |
> | FMNIST/LeNet          | 46.30 ± 7.08   | 9.71 ± 0.06   |
> | SVHN/ResNet_small     | 304.16 ± 0.96  | 108.87 ± 1.90 |
> | CIFAR-10/ResNet_small | 177.08 ± 11.68 | 70.17 ± 0.88  |
>
>
> As shown by the table, KIVI takes from 2-5 times the training time of IVON, where aspects such as alternating projections (and whether the stochastic variant is used) and the number of MLE steps used for warmup can affect the total execution time.
>
> > The new method does not consistently outperform baselines in predictive and calibration performance. How are these findings explained? What makes SVHN a better benchmark to demonstrate the advantages of the methods compared to MNIST and Fashion MNIST?
>
> The takeaway from, e.g, Tables 1 and 2 is that KIVI outperforms methods like SWAG and Miani et al. consistently, and performs on par with IVON and last-layer Laplace across benchmarks, beating them on SVHN. We don't claim that SVHN is a better benchmark than MNIST or Fashion MNIST (if there are misleading statements in the paper, please let us know and we'll correct them!).
>
> Notably, we emphasize that on Accuracy, NLL, and MCE (two "quality-of-fit" and one calibration metric), our method consistently performs best or second-best, while this level of reliability is not observed in the baseline methods. We emphasize that consistency is one of the key challenges facing BNNs.
>
> > A discussion on limitations of extending the method towards language models would help the reader get a clearer understanding of the scope and generalizability of the work.
>
> Thank you for raising this crucial point. In principle, our method applies to large language models. Since we use implicit solvers to compute projections, our method has the same space complexity as standard optimization. While the projections introduce an additional computational overhead, the computational complexity is linear in parameter size, batch size, and the number of Monte Carlo samples. So training is only slower by a constant factor. Hence, it is feasible to use our method whenever it is feasible to train a model.
>
> Thanks again for the thorough review! We hope that all concerns have been addressed adequately, and look forward to the discussion.

---

### Note · Authors · 2025-08-12

We thank all reviewers for their time, the positive evaluation of our paper, and the helpful discussions. Based on the feedback, the submission now includes clearer notation in the technical parts, a discussion of computational complexity (and pseudocode), and we'll also include SGMCMC and ensemble baselines to strengthen the experimental evidence. We hope that thereby, all concerns have been addressed adequately.


Thanks again!


Best wishes,

Author(s)

---

### Decision · Program_Chairs · 2025-09-17

**Decision:**

Accept (poster)

**Comment:**

This paper introduces KIVI, a variational inference method for Bayesian neural networks that decomposes the posterior covariance into two subspaces based on the kernel and image of the network's Jacobian.

The final reviews (post-discussion) show strong support for acceptance, with all reviews but one recommending acceptance. The reviewers in favor of acceptance praise the paper's originality. Reviewer bAN4 calls it one of the most exciting papers in the field. The work is viewed as a well-executed and valuable contribution that brings fresh ideas to BNN research. The primary dissenting opinion from reviewer rdDS centers on the need for additional baselines and a better discussion of prior work. They argue that given the method's computational cost, a comparison against non-VI methods like SGMCMC or deep ensembles is needed. I agree with reviewer rdDS and expect the updated manuscript to address these issues.

Two concerns that have not been raised by reviewers are the timeliness of this work and the experimental execution. While I agree with the reviewers that the submission introduces interesting ideas, I believe the experimental results are far below what I would expect to see in a Bayesian deep learning paper in 2025. The paper only considers toy and computer vision experiments, and the results (in terms of accuracy) are **significantly** below what would be expected in the deep learning literature, even a few years ago. I suspect the low predictive accuracy is due to the choice of model architecture (ResNet34 instead of ResNet18 or ResNet50) and other implementation decisions that are orthogonal to the proposed method. However, without results that show that the proposed methods outperforms MAP baselines that are on par with strong results reported in the deep learning literature (e.g., a 95% test accuracy on CIFAR-10 with a ResNet18 trained from scratch), this work is unlikely to have impact beyond the BNNs community. I strongly recommend to the authors to include such results in the updated manuscript. For reference points in the Bayesian deep learning literature, see for example https://arxiv.org/abs/2311.15990 and https://arxiv.org/pdf/2309.09814.

Despite these concerns, I believe the paper has merit and would be of interest to the NeurIPS community. I recommend acceptance.